# Wind turbine load validation in wakes using wind field reconstruction techniques and nacelle lidar wind retrievals

Davide Conti[1], Vasilis Pettas[2], Nikolay Dimitrov[1], and Alfredo Peña[1]

[1]Department of Wind Energy, Technical University of Denmark, Frederiksborgvej 399, 4000 Roskilde, Denmark
[2]Stuttgart Wind Energy (SWE), University of Stuttgart, Allmandring 5b, 70569 Stuttgart, Germany

**Correspondence:** Davide Conti (davcon@dtu.dk)

**Abstract.** This study proposes two methodologies for improving the accuracy of wind turbine load assessment under wake conditions by combining nacelle-mounted lidar measurements with wake wind field reconstruction techniques. The first approach consists of incorporating wind measurements of the wake flow field, obtained from nacelle lidars, into random, homogeneous Gaussian turbulence fields generated using the Mann spectral tensor model. The second approach imposes wake deficit time-series, which are derived by fitting a bivariate Gaussian shape function on lidar observations of the wake field, on the Mann turbulence fields. The two approaches are numerically evaluated using a virtual lidar simulator, which scans the wake flow fields generated with the Dynamic Wake Meandering (DWM) model, i.e., the *target* fields. The lidar-reconstructed wake fields are then input to aeroelastic simulations of the DTU 10 MW wind turbine for carrying out the load validation analysis. The power and load time-series, predicted with lidar-reconstructed fields, exhibit a high correlation with the corresponding *target* simulations; thus, reducing the statistical uncertainty (realization-to-realization) inherent to engineering wake models such as the DWM model. We quantify a reduction in power and loads' statistical uncertainty by a factor between 1.2 and 5, depending on the wind turbine component, when using lidar-reconstructed fields compared to the DWM model results. Finally, we show that the amount of lidar-scanned points in the inflow and the size of the lidar probe volume are critical aspects for the accuracy of the reconstructed wake fields, power, and load predictions.

## 1 Introduction

Wind turbines operating under wake conditions experience higher loading and lower power productions than those operating under wake-free conditions (Barthelmie et al., 2009; Larsen et al., 2013). The wake-induced velocity deficit and its meandering are critical aspects in both loads and power analyses (Madsen et al., 2010; Doubrawa et al., 2017). The former reduces the inflow wind speed and causes unbalanced aerodynamic load distribution at the rotor, which in turn induces high load cycle amplitudes in the whole wind turbine structure (Lee et al., 2012). The latter is the main source of wake added turbulence (Madsen et al., 2010), affecting wind turbine responses and inducing high fatigue damage (Larsen et al., 2013). Moreover, small turbulence eddies that result from the breakdown of the tip vortices can cause small fatigue load cycles (Madsen et al., 2005). Thus, aeroelastic analysis of wind turbines operating under wake conditions requires detailed modeling of the wake flow fields.

To date, detailed predictions of wake-generated turbulence can be achieved with large eddy simulation (LES); however, the computational cost is prohibitive when large number of simulations are required. This makes engineering wake models a practical alternative for certain applications. For design load evaluation, the IEC 61400-1 standard (IEC, 2019) recommends the Dynamic Wake Meandering (DWM) model, among other low-order engineering wake models.

The DWM model considers wakes to act as passive tracers displaced in the lateral and vertical directions by the large eddies of the atmospheric flow (Madsen et al., 2010). The wake field is modelled as a 'cascade' of quasi-steady velocity deficits emitted by the source turbine that meander through a pre-calculated stochastic meandering path and that are advected in the stream-wise direction adopting Taylor's hypothesis of frozen turbulence. These wake deficit time series are superposed on random three-dimensional turbulence fields serving as input for aeroelastic simulations (Larsen et al., 2008; Madsen et al., 2010). The wake flow features simulated by the DWM model are conditional on both the ambient conditions, which can be measured from a local meteorological mast, and the operational conditions of the upstream wind turbines. In order to carry out load simulations, the 10-min statistical properties (mean and variance) of the simulated ambient inflow are set to match the measured ambient wind statistics (Dimitrov and Natarajan, 2017).

There are three primary sources of uncertainty intrinsic of engineering wake models that affect the accuracy in power and load predictions, which we here denote as the measurement, modeling and statistical uncertainty. The measurement uncertainty includes deviations between the measured quantity of interest (e.g., the ambient wind field's characteristics or the power and load data) and their actual true values. The modeling uncertainty originates from the simplistic flow modeling assumptions adopted to describe wake flow fields. This type of uncertainty can partly be reduced by improving the wake model (e.g., by adding further physical effects) (Keck et al., 2015) or by calibrating model parameters using measurements (Larsen et al., 2013; Reinwardt et al., 2020). Calibrating the DWM model with site-specific observations improves the accuracy in power and load estimates; however, such calibrations do not hold at other sites (Madsen et al., 2010; Keck et al., 2012; Larsen et al., 2013; Reinwardt et al., 2020). As a result, DWM model-based power and load assessments might be highly uncertain at a given site unless high spatial and temporal resolution measurements of the wake are available for model calibration.

The statistical uncertainty derives from the traditional method of performing aeroelastic simulations, for which the numerical wind fields are set to match the statistical properties (mean and variance) of the observed wind field on a 10-min basis. Since the numerical turbulence field and the wake meandering are stochastic processes, the instantaneous velocities of the simulated wake wind field and the resulting load prediction time-series are uncorrelated with the observations. This can lead to simulation errors (Zwick and Muskulus, 2015) and introduces high statistical uncertainty on power and load predictions (Dimitrov and Natarajan, 2017; Pedersen et al., 2019). Further, to accurately predict power and loads in a load validation analysis, it is essential to accurately reconstruct wake meandering time series.

Alternative load verification procedures are being explored to potentially reduce the statistical and modelling uncertainty of engineering wake models and replace measurements from masts with those from Doppler lidars (Dimitrov et al., 2019; Reinwardt et al., 2020; Conti et al., 2020). Lidars can provide high spatial and temporal resolution inflow observations and extend (and eventually replace) traditional point-like measurements such as those from cup and sonic anemometers. Further, as modern wind turbines have considerably increased in size, reaching rotor diameters of the order of 150–200 m, accurate

measurements of the inflow wind field for aeroelastic calculations require multi-point and multi-height wind measurements within the entire rotor plane.

In particular, nacelle-mounted lidars have the advantage of being aligned with the rotor, which increases the amount of validation data in contrast to a fixed mast where only a small wind direction sector is valid. The feasibility of nacelle-mounted lidar observations has been demonstrated for wake characterization (Trujillo et al., 2011; Fuertes et al., 2018; Herges and Keyantuo, 2019; Reinwardt et al., 2020), lidar-assisted control (Schlipf et al., 2013; Simley et al., 2013, 2018), and power and load analysis in free-stream conditions (Wagner et al., 2014; Dimitrov et al., 2019).

The recent work of Conti et al. (2020) proposed a lidar-based load validation procedure under wake conditions that describes wake flow fields by means of time-averaged wind field characteristics estimated using nacelle lidar measurements. Although the quantified uncertainty in lidar-based power and load predictions was found comparable to estimates from IEC-recommended practices that uses the DWM model (Conti et al., 2020), the authors stated that lidar-based load validation procedures in wakes should account for a model of the wake deficit and its meandering dynamics to predict power and loads accurately.

Overall, developing lidar-based wake wind field reconstruction techniques that reduce the modeling and statistical uncertainties in the inflow inherent of low-order engineering wake models can improve loads and lifetime estimations accuracy (Rommel et al., 2020), enhance power curve testing in wind farms (Lydia et al., 2014; Wagner et al., 2015), and promote lidar-assisted wind turbine and wind farm control strategies (Bossanyi et al., 2014; Raach et al., 2017; Simley et al., 2018; Schlipf et al., 2020).

The present work proposes two alternative approaches for wind turbine load validation under wake conditions using nacelle-mounted lidar measurements combined with wake wind field reconstruction techniques. The first approach builds on the work of Dimitrov and Natarajan (2017), which incorporates multiple lidar retrievals in a turbulence field generated using the Mann spectral model (Mann, 1994) through a constrained Gaussian field algorithm. Incorporating nacelle-lidar measurements as constraints into turbulence fields can circumvent the DWM model's assumption to consider wakes as passive tracers (Madsen et al., 2010), while reconstructing the actual observed inflow at a high spatial and temporal resolution.

The second approach reconstructs wake deficit characteristics including wake meandering by fitting a bivariate Gaussian shape function on lidar retrievals and superimposes these deficits on a random realization of the Mann turbulence field. This approach intends to minimize errors in wake deficit representations and introduce the observed wake meandering path directly in the simulations. Both lidar-based wake field reconstruction techniques can potentially decrease the modeling and statistical uncertainty inherent to the DWM model, thus predicting accurate power productions and loads.

We evaluate these lidar-based wake field reconstruction techniques on a tailored-designed numerical framework that simulates a nacelle-mounted lidar scanning the synthetic wake fields generated with the DWM model. The main objective of this study is to verify that nacelle-mounted lidar measurements incorporated into wake field reconstruction methods improve the accuracy of power and load predictions when compared to wake field reconstruction using engineering wake models alone.

The work is structured as follows. In Sect. 2, we briefly formulate the load validation procedure. Section 3 introduces the methodology including the Mann spectral tensor model (Sect. 3.1) and the DWM model (Sect. 3.2). Section 3.3 describes the virtual lidar simulator and the analyzed scanning configurations. The wake field reconstruction techniques are formulated in

Sect. 3.4. The results are provided in Sect. 4, including the uncertainty analysis of the lidar-reconstructed fields in Sect. 4.1, a
detailed analysis of the load validation results in Sect. 4.2, and the sensitivities of the lidar specifications, e.g., probe volume
size and sampling frequency, and those related to the atmospheric inflow conditions on the load predictions accuracy in Sect.
4.3. The last two sections are dedicated to the discussion of the findings and the conclusions from the study.

## 2   Problem formulation

The design load cases (DLCs) and load verification procedure for wind turbines operating in wakes are described in the IEC
standards (IEC, 2015, 2019). The present work covers the analysis of fatigue loads of wind turbines operating in wakes (see
IEC 61400-1, DLC1.2). We apply the one-to-one load validation procedure of the IEC 61400-13 (IEC, 2015), which consists
of comparing simulated and targeted (e.g., measured) load statistics to assess the accuracy of aeroelastic simulations. As we
carry out the load validation analysis numerically, we define a tailored-designed load validation procedure, inspired by the
approach of Dimitrov and Natarajan (2017) and illustrated in Fig. 1. The DTU 10 MW wind turbine (Bak et al., 2013) is used
as reference in this study.

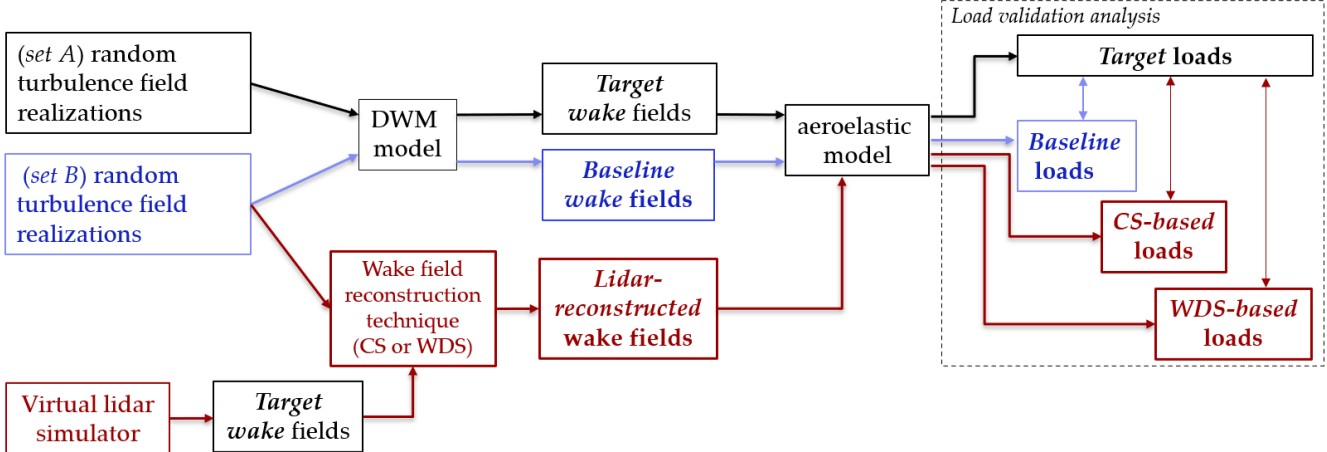

**Figure 1.** An illustration of the numerical framework utilized to reconstruct wake fields through the DWM model and our proposed lidar-
based wake field reconstruction techniques (i.e., the constrained simulations CS and the wake deficit simulations WDS). Further, this frame-
work allows quantifying the uncertainty in power and load predictions resulting from aeroelastic simulations with the DWM model-based
and lidar-based wake fields. More details can be found in the text.

We use two sets of random turbulence field realizations, which we denote as *set A* and *set B*. These turbulence fields are
generated using the model by Mann (1994); thus, they are defined as zero-mean, homogeneous, uniform-variance Gaussian
random fields. We simulate DWM model-based wake fields using turbulence realizations from *set A*, which we denote as the
*target* fields (see the black rectangular box in Fig. 1).

In contrast, the DWM model-based wake fields using turbulence field realizations from *set B* are denoted as the *baseline* (see the blue rectangular box in Fig. 1). Since the turbulence fields from *set A* and *set B* have the same turbulence characteristics, as they are generated using the same Mann parameters but are statistically independent (i.e., the resulting wind fields time series are uncorrelated), we expect that the outcomes of load simulations with *set A* and *set B* will have the same statistical properties but will not be correlated (Dimitrov and Natarajan, 2017).

Hence, the result of a one-to-one comparison of load statistics between the *baseline* and the *target* simulations is a direct measure of the statistical uncertainty (i.e., load scatter) that originates from both the random Mann-based turbulence realizations and the stochastic meandering process inherent to the DWM model. In a traditional load validation analysis, the *target* loads will be the measured loads, whereas the *baseline* loads will be the loads resulting from aeroelastic simulations using turbulence fields with the same properties as the measured inflow conditions (IEC, 2015).

To evaluate the lidar-based approaches, we use a virtual lidar simulator that scans the *target* wake fields, and, through our proposed wake field reconstruction technique, incorporates these samples in a random turbulence field realization from *set B* (see Fig. 1). This numerical approach intends to imitate what we would eventually do when nacelle lidar measurements within wakes are available for load predictions.

Further, by incorporating lidar retrievals in the wind field reconstruction technique, we expect to reduce the amount of statistical uncertainty as the load time series resulting from this approach will have greater similarity with the load time series based on the *target* turbulence fields. Therefore, this procedure allows us to quantify the uncertainty of load predictions that results from lidar-reconstructed wake fields (see the red elements in Fig. 1) against the *target*, and at the same time, to compare the associated statistical uncertainty with that of the *baseline*. To summarize, the following load simulation cases are defined:

- *Target*: DWM model-based wake fields imposed on random turbulence field realizations from *set A*.

- *Baseline*: DWM model-based wake fields imposed on random turbulence field realizations from *set B*.

- *Constrained simulations (CS)*: lidar-reconstructed wake fields, where lidar virtual measurements of the *target* fields are incorporated as constraints to random turbulence field realizations from *set B*.

- *Wake deficit simulations (WDS)*: lidar-reconstructed wake fields, where lidar virtual measurements of the *target* fields are fitted to a wake deficit shape function to compute wake deficits, which are then superimposed to random turbulence field realizations from *set B*.

The load validation comprises a large number of simulations (we use eighteen random turbulence field realizations for each individual 10-min statistic of the inflow wind), to quantify the statistical uncertainty of power and load predictions under inflow conditions measured at a site. More details on the load validation analysis are provided in Sect. 4.2. Eventually, we quantify the load uncertainties of the *baseline*, *CS*- and *WDS*-methods by comparison to the loads of the *target* simulations, and we define two main criteria to evaluate the proposed approaches:

I The bias (here defined as the mean ratio between simulated and *target* loads) obtained with the lidar-reconstructed *CS*- and *WDS*-simulations is equal to that obtained with the *baseline*.

II  The statistical uncertainty (here defined as the standard deviation of the ratio between simulated and *target* loads) derived with the lidar-reconstructed *CS*- and *WDS*-simulations is lower than that obtained with the *baseline*.

Provided that these criteria are satisfied, the proposed lidar-based wake field reconstruction techniques will produce (I) power and load predictions in wakes that are statistically unbiased compared to the DWM model results, and (II) a reduced statistical uncertainty in power and load predictions compared to the DWM model results, which is achieved by reconstructing wake fields with stronger similarities to the actual inflow.

## 3  Methodology

### 3.1  Mann turbulence spectral model

The time-domain aeroelastic simulations require input of a three-dimensional turbulence field that mimics atmospheric turbulence (Dimitrov et al., 2017). For this purpose, the IEC 61400-1 recommends, i.a., the Mann uniform shear spectral tensor model (Mann, 1994) or the Kaimal model (Kaimal et al., 1972). The turbulence spectral properties of a three-dimensional homogeneous wind field are described by the spectral velocity tensor $\Phi_{ij}(\boldsymbol{k})$ (Kristensen et al., 1989):

$$\Phi_{ij}(\boldsymbol{k}) = \frac{1}{(2\pi)^3} \int R_{ij}(\boldsymbol{r})\exp(-\mathrm{i}\boldsymbol{k}\cdot\boldsymbol{r})d\boldsymbol{r}, \tag{1}$$

which is the Fourier transform of the covariance tensor $R_{ij}(\boldsymbol{r})$, $\boldsymbol{r} = (x, y, z)$ is the spatial separation vector defined in a right-handed coordinate system such that the longitudinal component of the wind field ($u$) is in the $x$ direction, $y$ and $z$ are the directions of the transverse components (i.e., the $v$- and $w$-velocity components), and $\boldsymbol{k} = (k_1, k_2, k_3)$ is the vector with the wavenumbers in the $(x, y, z)$ directions.

The model by Mann (1994) (hereafter referred to as the Mann model), assumes neutral atmospheric conditions and defines the spectral tensor as function of three input parameters: $\alpha_k\epsilon^{2/3}$, which is a product of the spectral Kolmogorov constant $\alpha_k$ and the turbulent energy dissipation rate $\epsilon$, $\Gamma$ is a parameter describing the anisotropy of the turbulence, and $L$ is a length scale proportional to the size of turbulence eddies. From the spectral tensor, the cross-spectra between two points located in a $y$–$z$ plane and separated by a distance $(\Delta_y, \Delta_z)$ are calculated numerically by:

$$\chi_{ij}(k_1, \Delta_y, \Delta_z) = \int\int \Phi_{ij}(\boldsymbol{k}, \alpha_k\epsilon^{2/3}, L, \Gamma)\exp(\mathrm{i}k_2\Delta_y + \mathrm{i}k_3\Delta_z)dk_2 dk_3. \tag{2}$$

Further, by inverse Fourier-transforming the cross spectrum $\chi_{ij}$, we can derive the auto- and cross-correlation structure of the turbulence field (Dimitrov and Natarajan, 2017), as:

$$R_{ij}(\Delta_x, \Delta_y, \Delta_z) \propto \int \chi_{ij}(k_1, \Delta_y, \Delta_z)\exp(\mathrm{i}k_1\Delta_x)dk_1. \tag{3}$$

### 3.2  Dynamic Wake Meandering model

The DWM model is an engineering wake model that simulates wind field time series and includes three components: a quasi-steady velocity deficit, the wake-added turbulence, and the wake meandering (Madsen et al., 2010). Figure 2 illustrates these wake features components qualitatively.

The DWM model assumes wakes as passive tracers displaced in the lateral and vertical directions by the large eddies in the atmospheric flow. Further, the quasi-steady wake deficits are advected in the stream-wise direction adopting Taylor's assumption of frozen turbulence (Madsen et al., 2010). This set of assumptions allows decoupling the wake deficit and wake-added turbulence components from the wake meandering model (Larsen et al., 2007). Hence, the three components of the DWM model are computed separately and subsequently superposed on random homogeneous turbulence field realizations (e.g., generated using the Mann model) to produce three-dimensional wake field time series that are input to aeroelastic simulations (Larsen et al., 2013; Keck et al., 2014).

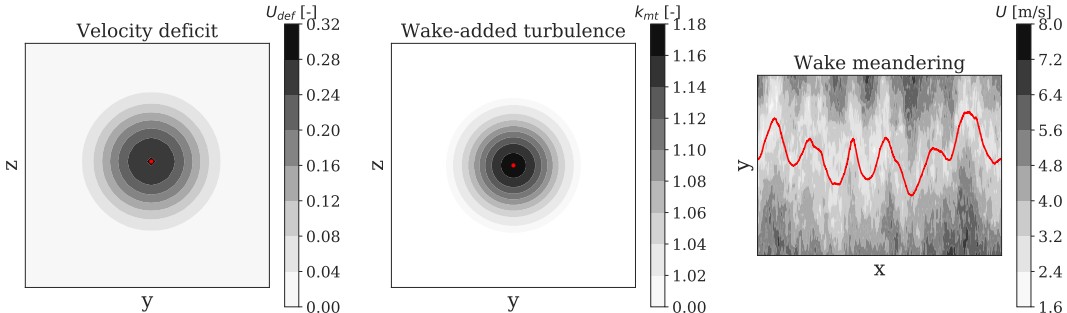

**Figure 2.** Qualitative representation of the three wake components predicted by the DWM model, including an axisymmetric quasi-steady velocity deficit, which is defined as the local wind speed $U$ divided by the ambient wind speed $\bar{U}_{amb}$ (left), a wake-added turbulence scaling factor, $k_{mt}$ (middle) that assumes zero values outside of the wake region, and the meandering of the quasi-steady wake deficit superposed on a random homogeneous turbulence field realization (right). The red marker identifies the wake center position and the red solid line the wake center's trajectory in the longitudinal $x$- and lateral $y$-coordinate. The wake also meanders in the vertical direction (not shown). The wake features are computed for an ambient inflow characterized by $\bar{U}_{amb}=$ 6 m/s and a turbulence intensity of $TI_{amb}=8\%$.

The velocity deficit definition is based on the work of Ainslie (1986, 1988), who applied a thin shear-layer approximation of the Navier–Stokes equations and a simple eddy viscosity formulation. The wake deficit expansion and recovery downstream of the generating turbine is driven by the turbulent mixing occurring due to the ambient turbulence and the turbulence generated by the wake shear field itself (Madsen et al., 2010; Keck et al., 2014, 2015). For a given wind turbine aerodynamic rotor design, a 10-min average inflow wind speed ($\bar{U}_{amb}$), and ambient turbulence intensity ($TI_{amb}$), the DWM model calculates a two-dimensional quasi-steady velocity deficit defined in the meandering frame of reference (MFoR), which is a coordinate system with origin in the center of symmetry of the deficit, as shown in Fig. 2-left. Here, we use the numerical scheme of the standalone DWM model (Liew et al., 2020; Larsen et al., 2020) to compute the quasi-steady velocity deficit.

The wake-added turbulence originating from the breakdown of tip vortices and from the shear of the velocity deficit is accounted for by a semi-empirical turbulence scaling factor (Madsen et al., 2010) as:

$$k_{mt}(y,z) = \mid 1 - U_{def}(y,z) \mid k_{m1} + \left| \frac{\partial^2 U_{def}(y,z)}{\partial y \partial z} \right| k_{m2}, \tag{4}$$

where $U_{def}$ is the axisymmetric velocity deficit in the MFoR (see also Fig. 2-left), and $k_{m1}$ and $k_{m2}$ are calibration constants (Madsen et al., 2010). The two-dimensional spatial distribution of $k_{mt}$ is shown in Fig. 2-middle. As wake turbulence is both highly isotropic and characterized by a reduced turbulence length scale compared that of the ambient turbulence (Madsen et al., 2005), $k_{mt}$ of Eq. (4) scales the residual field of a Mann-generated turbulence field with a standard deviation of the longitudinal wind component equal to 1 m/s (IEC, 2019), assuming isotropic turbulence, i.e., $\Gamma = 0$, and a small turbulence length scale ($L \approx$ 10–25% of the ambient turbulence length scale) (Madsen et al., 2010).

The wake meandering is assumed to be governed by the atmospheric turbulent structures of the order of two rotor diameters ($D$) or larger (Madsen et al., 2010). This assumption was verified using lidar observations of wakes (Bingöl et al., 2010; Trujillo et al., 2011). Thus, the simulated wake meandering time series is obtained by low-pass filtering atmospheric turbulence fluctuations (i.e., $v$- and $w$-velocity components measured from a local mast or lidar, or alternatively simulated by the Mann model) by a cut-off frequency $f_{cut,off} = \bar{U}_{amb}/(2D)$, which excludes contributions from smaller eddies to the meandering dynamics (Larsen et al., 2008).

As a result, the wake field simulated by the DWM model can be seen as a 'cascade' of quasi-steady velocity deficits that meander in the lateral and vertical directions and are advected downstream by the mean wind speed of the inflow using Taylor's assumption. These wake features are superposed on stochastic homogeneous turbulence field realizations to generate wake fields time-series that are then input to aeroelastic simulations (see Fig. 2-right).

Mathematically, a three-dimensional synthetic wake flow field compliant with the DWM model formulation can be defined by a linear superposition of the ambient wind field and two inhomogeneous turbulence terms as:

$$U_{\mathrm{DWM}}(x,y,z) = \bar{U}_{amb}(z) + u'_{i,K_{def}}(x,y,z) + u'_{j,K_{turb}}(x,y,z), \tag{5}$$

where $\bar{U}_{amb}(z)$ is the ambient wind speed including the atmospheric vertical wind shear profile, $u'_{iK_{def}}(x,y,z)$ is a residual turbulence field with imposed wake deficits that follow the meandering path, and $u'_{j,K_{turb}}$ is a second turbulence field modelling wake-added turbulence effects. Adopting Taylor's assumption, the wake field can be described by the spatial vector solely; thus, the time variable is disregarded in Eq. (5). The subscripts $i, j$ indicate two random and uncorrelated turbulence field realizations. The $u'_{i,K_{def}}$ field is computed as:

$$u'_{i,K_{def}}(x,y,z) = \bar{U}_{amb}(z)(1 - K_{def}(x,y,z)) + u'_i(x,y,z) - \bar{U}_{amb}(z), \tag{6}$$

where $K_{def}(x,y,z)$ denotes the DWM model-based wake deficit time-series including a pre-computed stochastic meandering path, and $u'_i$ is a random homogeneous turbulence field realization from the Mann model with the same Mann parameters as those of the ambient wind field. Note that $K_{def}(x,y,z)$ assumes values equal to unity when wake losses are not present. The Mann parameters, $\alpha_k \epsilon^{2/3}$, $L$, and $\Gamma$, are derived, e.g., from fitting the free-stream observed turbulence velocity spectra with the Mann model with the use of pre-computed look-up-tables to speed-up the fitting procedure (Peña et al., 2017). The wind field formulation of Eqs. (5) and (6) is consistent with the domain of wind fields typically input to aeroelastic simulations (Larsen and Hansen, 2007). Finally, $u'_{j,K_{turb}}$ is obtained as:

$$u'_{j,K_{turb}}(x,y,z) = u'_j(x,y,z)K_{mt}(x,y,z), \tag{7}$$

where $K_{mt}(x,y,z)$ denotes a time-series of turbulence's scaling factors computed from Eq. (4) including a pre-computed stochastic meandering path, $u'_j$ is a random homogeneous turbulence field with $\sigma_u$ = 1 m/s, $\Gamma$=0 and $L$= 10% of the ambient turbulence length scale (Madsen et al., 2010; IEC, 2019).

### 3.3 Lidar simulator

We use the lidar simulator developed within the ViConDAR open-source numerical framework to virtually replicate lidar measurements (https://github.com/SWE-UniStuttgart/ViConDAR), (Pettas et al., 2020). The lidar simulator derives the line-of-sight (LOS) velocities at each scanning location by transforming the $u$-, $v$- and $w$-velocity components of the synthetic turbulence field into a LOS coordinate system. To simulate the probe volume of the lidars, a Gaussian weighting function $W(F,r)$ is imposed along the LOS coordinate $r$ and centered at the focal distance $F$:

$$V_{LOS,eq} = \int V_{LOS}(r)W(F,r)dr. \tag{8}$$

The $u$-velocity is computed from the projection of $V_{LOS,eq}$ onto the longitudinal axis, i.e., the $v$- and $w$-velocity components are neglected in the field reconstruction (Schlipf et al., 2013; Simley et al., 2013). This assumption leads to:

$$u_{lidar} = \frac{V_{LOS,eq}}{\cos\phi\cos\theta}, \tag{9}$$

where $\phi$ is the elevation and $\theta$ the azimuth angle of the scanning pattern, which refer to the rotations about the $y$ and $z$ axes, respectively. Neglecting the $v$- and $w$-velocity components introduces uncertainty in the wind field reconstruction. However, the opening angles $(\phi,\theta)$ relative to the scanning configurations of our work reach a maximum of 35° (see Sect. 3.3.1); thus, the introduced errors by Eq. (9) are marginal (Simley et al., 2013).

Other sources of uncertainty in the radial velocity estimation inherent to lidars, e.g., from the optics and internal signal processing, are accounted for by adding a Gaussian white noise. Here we add noise at a level that results in a signal-to-noise ratio of $-20$ dB as in Pettas et al. (2020). We do not investigate the sensitivity of the noise level in the present work.

The lidar simulator can mimic any arbitrary scanning pattern and includes a time-lag between each lidar-sampled measurement to resemble the scanning frequency (see Fig. 3). In the present study, the virtual lidar data are computed from the synthetic wake flow fields generated using the DWM model. These wind fields are time series of the $u$-, $v$- and $w$-velocity components defined over a turbulence box with a grid size of 8192×32×32 ($x \times y \times z$). A spatial resolution of 6.5 m is used for the grid in the rotor plane, which leads to a turbulence box with dimension 208 m × 208 m in both lateral and vertical directions ($y \times z$). The spatial resolution $dx$ in the $x$-axis depends on the simulated ambient wind speed at hub height, $dx = (\bar{U}_{amb}T_{sim})/8192$, where $T_{sim}$ is the simulation time in seconds. These dimensions ensure an adequate turbulence field for a 10-min wind field simulation over a large rotor and a space-time resolution such that the probe volume effects can be captured by the virtual lidar (Dimitrov et al., 2017; Pettas et al., 2020).

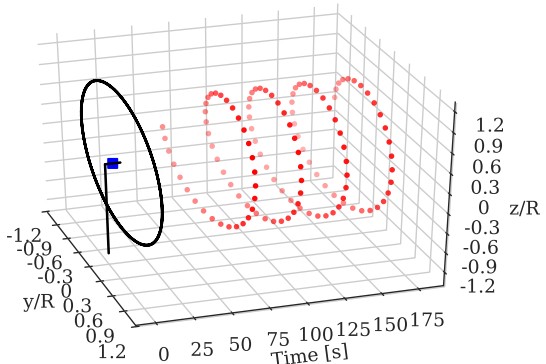

**Figure 3.** An illustration of the virtual lidar simulator setup run for 175 s with simulated time lag. The wind turbine is sketched by the black solid lines, the nacelle-mounted lidar is represented by a blue squared marker measuring upfront the turbine. The trajectory of the scanning beam is shown by discrete red dots.

### 3.3.1 Lidar scanning strategies

To evaluate currently available nacelle lidars' ability to perform wake characterization, we select a few standard scanning configurations and use them to perform load validation within wakes. These are a 4-beam lidar (4P) (Held and Mann, 2019a, b), an extended configuration with 7 beams, six arranged at the corner of a hexagon and a central beam (7P) (Pettas et al., 2020), the conical scanning lidar (Cone) (Medley et al., 2014; Borraccino et al., 2017; Peña et al., 2017), the SpinnerLidar (SL) (Peña et al., 2019; Doubrawa et al., 2019), and a general grid pattern (Grid) covering the full turbulence box (see Fig. 4).

The lidar simulator is assumed to scan the selected patterns at the same single range upwind of the rotor. Pulsed and continuous-wave (CW) lidar technologies apply different approaches at scanning multiple ranges (Peña et al., 2017). Pulsed lidars can scan multiple ranges along the LOS simultaneously within a single sample, while CW lidars typically sample much faster at a given range but need to refocus in order to change the sampling range. In the present paper, we only consider a single focusing range that is achievable with both lidar technologies. Further, a time lag between each sampling beam is simulated to mimic lidars' sampling frequency.

Although we do not optimize the scanning patterns, we use scan radii (defined as the radius between hub height and the location of the scanned points) of about 70–80% of the rotor radius to estimate wind field characteristics based on previous recommendations (Dimitrov and Natarajan, 2017; Simley et al., 2018). Thus, we define the 4P, 7P, and Cone patterns accordingly as shown in Fig. 4. The SL trajectory is scaled to cover the full rotor area, and the positions over the plane of measurement are separated by 29 m in both vertical and transverse directions for the Grid pattern.

A preview distance of 0.7D ($\approx$ 125 m) is assumed. Note that increasing the preview distance reduces the errors caused by the cross-contamination effects of the $v$- and $w$-components and reduces the induction effects, but raises errors due to the wind

evolution (Simley et al., 2012). These effects are not investigated in detail in this work, as we use DWM model-based wake fields as *target*, which do not include induction effects nor turbulence evolution as the Taylor's assumption is applied.

We assume a 2-s scan-period for all the simulated configurations, which refers to the time required for a beam to complete the full pattern. Given the finite resolution of the synthetic turbulence boxes (i.e., 6.5 m in both lateral and vertical directions), the Cone and SL scanned locations are binned within the box grid, as reported in Table 1.

A probe volume with an extension of 30 m in the LOS direction is assumed for all the analyzed patterns, which is comparable with the current CW lidar technology measuring at distances beyond 120 m (Peña et al., 2015). Further, a 30 m probe length is commonly used to model pulsed lidars (Schlipf, 2016). Here, we define the probe volume's length as the standard deviation of the Gaussian weighting function for convenience. Typically, Gaussian weighting functions are used to model pulsed lidars, whereas Lorentzian functions are used for CW lidars (Mann et al., 2010). Additionally, we define a case (Grid*) that neglects probe volume averaging effects (see Table 1).

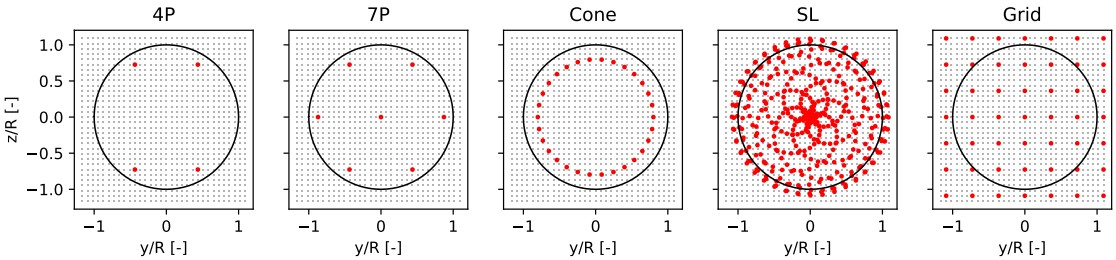

**Figure 4.** Selected lidar scanning patterns for the load analysis. The red markers indicate the scanned locations and the black dots in the background define the spatial resolution of the turbulence box. The rotor diameter is shown in a black solid line.

### 3.4 Wake field reconstruction techniques

By defining the DWM model-based wake flow fields as the *target* fields, the underlying assumptions on which we define the lidar-based wake field reconstruction techniques are:

1. The ambient wind conditions are known, including $\bar{U}_{amb}(z)$, the atmospheric turbulence intensity $(TI_{amb})$, and the atmospheric stability conditions (here implicitly prescribed through the Mann parameters: $\alpha_k\epsilon^{2/3}, L, \Gamma$).

2. The lidar-based wake fields are reconstructed by incorporating lidar observations (e.g., in the form of constraints or lidar-fitted velocity deficits) into a zero-mean, homogeneous, and random Gaussian turbulence field generated by the Mann spectral tensor model.

**Table 1.** Technical properties of the simulated lidar scanning configurations. Note that the Cone and SL measurements are binned according to the spatial resolution of the synthetic turbulence fields, thus leading to a reduction of the simulated scanning positions.

| Scanning configuration | Measurements / scan (binned) [-] | Sampling frequency [Hz] | Scan period [s] | Measurements / 10-min [-] | Probe volume size [m] |
|---|---|---|---|---|---|
| 4P | 4 | 2 | 2 | 1200 | 30 |
| 7P | 7 | 3.5 | 2 | 2100 | 30 |
| Cone | 100 (30) | 50 | 2 | 9000 | 30 |
| SpinneLidar (SL) | 400 (93) | 200 | 2 | 27900 | 30 |
| Grid | 49 | 25 | 2 | 14700 | 30 |
| Grid* | 49 | 25 | 2 | 14700 | 0 |

3. The induction effects on lidar measurements are neglected and the Taylor's frozen turbulence hypothesis is assumed.

4. Only the $u$-velocity fluctuations are reconstructed from the lidar measurements of the *target* wake fields.

The corresponding random turbulence field realizations from *set A* (used for the *target* fields) and *set B* have similar spectral properties; however, these fields only describe the turbulence structures of the ambient wind field. The lidar measurements of the wake field, combined with the wake field reconstruction approach, should recover the whole information regarding the wake characteristics, including velocity deficits, wake-added turbulence, and meandering in lateral and vertical directions. Further, the first assumption is no longer needed if a second instrument is deployed at the site measuring the ambient conditions, for example, using a mast or a nacelle-mounted lidar (Borraccino et al., 2017; Peña et al., 2017).

The second and third assumptions are inherent in the modelling approach and limitations of the DWM model and other analytical wake models; however, in this study, the wake characteristics are extracted directly from the lidar observations rather than a physical-based deficit formulation. Eventually, wind turbine responses are mainly affected by the mean wind speed in the longitudinal direction ($u$-velocity) and its variance (Dimitrov et al., 2018), while the effects of the $v$- and $w$-turbulence are generally marginal (Dimitrov and Natarajan, 2017).

### 3.4.1 Constrained Gaussian field simulations

The algorithm for applying constraints on a zero-mean, homogeneous, and isotropic Gaussian random field was developed in Hoffman and Ribak (1991), and extended to Mann-generated turbulence fields for aeroelastic simulations in Nielsen et al. (2003) and Dimitrov and Natarajan (2017). The algorithm uses a set of constraints that are here derived from a virtual lidar simulator and an unconstrained random turbulence realization generated with the Mann spectral tensor model.

Following the notation in Dimitrov and Natarajan (2017), we denote $\tilde{g}(\boldsymbol{r})$, where $\boldsymbol{r} = (x, y, z)$ is the spatial separation vector, an unconstrained random turbulence realization. The spectral property of $\tilde{g}(\boldsymbol{r})$ at each discrete lateral and vertical separation of the turbulence box can be computed from the Mann model in Eq. (2), given a set of parameters $(\alpha_k \epsilon^{2/3}, L, \Gamma)$. We denote a set of constraints as $\boldsymbol{H} = \{h_i(\boldsymbol{r})|_{r_i} = c_i, i, ..., M\}$, where each constraint is a measured value of the wind speed for a particular

spatial location $\boldsymbol{r}$ and $M$ is the total number of constraints (i.e., the number of scanned points within a 10-min period). Note that the constraints are defined as a residual wind field; thus, we remove the mean ambient wind speeds from the lidar measurements of Eq. (9), i.e., $c_i(\boldsymbol{r}_i) = u_{lidar}(\boldsymbol{r}_i) - \bar{U}_{amb}(\boldsymbol{r}_i)$, which are the values that are input to the algorithm.

The objective of the algorithm is to define a turbulence field $g(\boldsymbol{r})$, subjected to the constraints in $\boldsymbol{H}$ that maintains the covariance and coherence properties of the unconstrained field $\tilde{g}(\boldsymbol{r})$. As demonstrated in Dimitrov and Natarajan (2017), the unknown points of the field can be defined by maximizing their conditional probability distribution on the constraint set $\boldsymbol{H}$. We define the residual field $\xi(\boldsymbol{r}) = g(\boldsymbol{r}) - \tilde{g}(\boldsymbol{r})$, which is the difference between the constrained and unconstrained fields. This residual field is also a random Gaussian field, where its values at the constraint locations are known $\xi(\boldsymbol{r_i}) = c_i(\boldsymbol{r_i}) - \tilde{g}_c(\boldsymbol{r_i})$. The values of the residual field at unknown locations can be derived as (Dimitrov and Natarajan, 2017):

$$\bar{\xi}(\boldsymbol{r}) = \langle \xi(\boldsymbol{r}) | \boldsymbol{H} \rangle = \boldsymbol{\zeta}(\boldsymbol{r}) \boldsymbol{Z}^{-1} (\boldsymbol{H} - \tilde{g}_c(\boldsymbol{r})), \tag{10}$$

where $\langle . \rangle$ denotes ensemble averaging, $\boldsymbol{\zeta}(\boldsymbol{r})$ is a vector of cross-correlations between the constraints and the field, and $\boldsymbol{Z}$ is the symmetric correlation matrix of the constraints set. Both $\boldsymbol{\zeta}(\boldsymbol{r})$ and $\boldsymbol{Z}$ can be computed from Eq. (3). Eventually, any constrained realization can be written as a sum of the unconstrained field and the mean of the residual field as:

$$g(\boldsymbol{r}) = \tilde{g}(\boldsymbol{r}) + \boldsymbol{\zeta}(\boldsymbol{r}) \boldsymbol{Z}^{-1} (\boldsymbol{H} - \tilde{g}_c(\boldsymbol{r})). \tag{11}$$

By denoting $u'_{\text{CS,B,i}} = g(\boldsymbol{r})$, as the constrained turbulence field that incorporates lidar measurements into a random turbulence realization $i$ from *set B* (see Fig. 1), we can derive the reconstructed wake flow field to be input in aeroelastic simulations as:

$$U_{\text{CS}}(x, y, z) = \bar{U}_{amb}(z) + u'_{\text{CS,B,i}}(x, y, z). \tag{12}$$

Note that the fidelity of the reconstructed wind field will depend on the accuracy of the nacelle lidar measurements used to characterize the wake field.

### 3.4.2 Wake deficit superposition simulations

The wake deficit superposition (*WDS*) approach assumes that velocity deficits can be described by a bivariate Gaussian shape function, which is fitted based on lidar measurements of the *target* wake flow field. Several studies have demonstrated the viability and robustness of the Gaussian curve fitting to track wake deficit displacements in the far-wake region (Trujillo et al., 2011; Reinwardt et al., 2020).

In our study, the wake shape function not only tracks the wake meandering, but it is used to quantify the depth and width of the wake at each quasi-instantaneous scan performed by the lidar. Traditionally, the normalized velocity deficit is defined as the difference between the ambient wind speed and that inside the wake as:

$$U_{def}(x, y, z) = \frac{\bar{U}_{amb}(z) - u_{lidar}(x, y, z)}{\bar{U}_{amb}(z)}, \tag{13}$$

where $\bar{U}_{amb}(z)$ is assumed to be known and the lidar measurements in the wake ($u_{lidar}$) are sampled by the lidar simulator using Eq. (9). Following the procedure of Trujillo et al. (2011), a bivariate Gaussian shape is used to describe the velocity deficit flow field as:

$$K_{def,Gau}(y,z) = \frac{A}{2\pi\sigma_{wy}\sigma_{wz}} \exp\left[-\frac{1}{2}\left(\frac{(y_i - \mu_y)^2}{\sigma_{wy}^2} + \frac{(z_i - \mu_z)^2}{\sigma_{wz}^2}\right)\right], \tag{14}$$

where $(\mu_y, \mu_z)$ define the wake center location, $(\sigma_{wy}, \sigma_{wz})$ are width parameters of the wake profile in the $y$ and $z$ directions, respectively, $(y_i, z_i)$ denote the spatial location of the LOS and $A$ is a scaling parameter dictating the depth of the wake. A least squares method is applied to fit the measured wind speed deficits from Eq. (13) to the bivariate Gaussian function in Eq. (14).

The optimal wake deficit parameters $(\mu_y, \mu_z, \sigma_{wy}, \sigma_{wz}, A)$ are obtained for each completed scanning period (i.e., $\sim 2$ s as described in Table 1), resulting in approximately 300 lidar-reconstructed deficits within a 10-min period. Finally, these lidar-fitted wake deficits are superimposed on a random homogeneous turbulence field realization from *set B*, as shown in Fig. 1.

A preliminary analysis showed that wide turbulence boxes (208 m $\times$ 208 m) can present large turbulence structures within, i.e., broad regions across the box characterized by low wind speeds, whose sizes can alter the depth and width properties of the lidar-fitted wake deficits in Eq. (14). As a result, the wake properties of the reconstructed field can considerably deviate from the actual imposed wake characteristics.

To compensate for these deviations and considering that the DWM model-based wake fields can be defined as a linear summation of the ambient wind field $\bar{U}_{amb}$ scaled by the wake deficit function $K_{def}$, and a random homogeneous turbulence realization term $u_i'$, as reported in Eq. (6), we reformulate the least squares minimization problem as:

$$\Gamma_{def} = \sum_{mn}\left[U_{def}(y_m, z_n) - \frac{\bar{U}_{amb} - (\bar{U}_{amb}(z)(1 - K_{def,Gau}(y_m, z_n | \mu_y, \mu_z, \sigma_{wy}, \sigma_{wz}, A)) + u_{B,i}'(y,z))}{\bar{U}_{amb}(z)}\right]^2, \tag{15}$$

where subscripts $(m,n)$ indicate data points within the scanning configuration, the second term in the right-hand side defines the velocity deficit as in Eq. (13), in which the reconstructed wake field is defined as $\bar{U}_{amb}(1 - K_{def,Gau}) + u_{B,i}'$ and $u_{B,i}'$ is the random homogeneous turbulence realization from *set B*. Note that when wake losses are present, $(1 - K_{def,Gau})$ will reduce the ambient wind speed, as expected. As the sampling frequency of the lidar is lower than the sampling frequency of the synthetic wind field, we interpolate the fitted wake characteristics at each scan to the whole turbulence field by applying a nearest-neighbor interpolation scheme. Finally, the reconstructed wake field input to aeroelastic simulations is defined by:

$$U_{\text{WDS}}(x,y,z) = \bar{U}_{amb}(z)(1 - K_{def,Gau}(x,y,z)) + u_{B,i}'(x,y,z), \tag{16}$$

where $K_{def,Gau}(x,y,z)$ is fitted using Eq. (15) for each completed scan by the nacelle lidar.

## 4 Results

The results are divided into three parts. First, we assess the accuracy of lidar-reconstructed wake fields against *target* fields in Sect. 4.1. Second, we carry out the load validation analysis in Sect. 4.2, and separately present the load prediction uncertainty of

the *CS*-approach in Sect. 4.2.2 and that of the *WDS*-approach in Sect. 4.2.3. A detailed analysis of the predicted load time-series and load spectral properties is conducted in Sects. 4.2.4 and 4.2.5. Finally, we evaluate the sensitivities of both atmospheric turbulence conditions and the selected lidar technical specifications on the load prediction accuracy in Sect. 4.3.

## 4.1 Uncertainty of reconstructed wake fields

In this section, we evaluate the accuracy of the lidar-reconstructed fields against the *target* fields. At first, we assess the accuracy of the reconstructed $u$-velocity time-series across the turbulence box, by computing the root mean square error, RMSE $= \sqrt{1/n \sum_i^n (\tilde{y}_i - \hat{y}_i)^2}/\bar{y}_i$, between the lidar-reconstructed ($\tilde{y}$) and *target* velocity ($\hat{y}$), where $n = 8192$ is the grid size of the box in the longitudinal direction, normalized over the mean *target* velocity ($\bar{y}_i$) at each grid point of the turbulence box. The normalized RMSE provides a measure of the quality of the lidar-reconstructed fields with respect to the *target* fields; values closer to zero indicate a high precision and accuracy (see Fig. 5-top row).

Further, we compute the explained variance ratio across the turbulence box $\rho_E^2 = (\text{cov}(\tilde{y}, \hat{y})/\sigma_{\tilde{y}}\sigma_{\hat{y}})^2$ (i.e., the square of the Pearson's correlation coefficient (Achen, 1982)), which defines the proportion of the variance in the inflow field that is transferred to the unconstrained turbulence field by imposing the constraints (Dimitrov and Natarajan, 2017). As the *target* and lidar-reconstructed fields are based on two sets of random uncorrelated turbulence field realizations (see *sets A* and *B* in Fig. 1), $\rho_E^2 \sim 0$ is expected across the box, if no lidar information was included. Contrarily, $\rho_E^2 = 1$ indicates that the reconstructed time-series is fully-correlated with the *target*, thus the two fields match completely.

Figure 5 also shows the spatial distribution of $\rho_E^2$ derived from the *CS*- and *WDS*-reconstructed fields, with the 7P, Cone, and Grid configurations (see Table 1 for specifications). For this particular analysis, the turbine of interest is located 5D downstream of the upstream turbine, where D = 179 m is the diameter of the DTU 10 MW turbine, and ambient conditions characterized by $U_{amb}$ = 6 m/s and $\text{TI}_{amb}$ = 8 %. The inflow wind profile is defined by a power-law model with a shear exponent of 0.2.

As shown in Fig. 5, the locations of the imposed constraints are characterized by the lowest RMSE and highest $\rho_E^2$. This effect is more pronounced for the *CS* results, as the algorithm imposes the actual observations directly in the synthetic field. The RMSE would tend to zero, if the length of probe volume is neglected, the lidar's sampling frequency corresponds to the sampling frequency of the wind field, and cross-contamination effects are compensated. The RMSE increases (and $\rho_E^2$ decreases) for spatial regions that are farther from the lidar's beams. This occurs due to the covariance structure of the unconstrained turbulence field, for which the unknown points are nearly uncorrelated with the imposed constraints.

The errors introduced by the *WDS*-fields are partly a consequence of an inaccurate estimation of the wake deficit characteristics (i.e., due to the limited spatial scanning configuration) and the small-scale turbulence structures contained in the turbulence box. Finally, the results in Fig. 5 confirm that the amount of scanned positions by the lidar has a significant impact on the reconstructed fields' accuracy affecting both the mean and variance of the reconstructed $u$-velocity component. Therefore, patterns that cover a larger region of the rotor lead to more accurate field representations (Dimitrov and Natarajan, 2017; Pettas et al., 2020).

In Fig. 6, we compare the lidar-reconstructed $u$-velocity time-series extracted at hub height, using the Grid pattern, with the *target* observations derived at the same location. The *target* wake field is simulated with $U_{amb}$ = 6 m/s and $\text{TI}_{amb}$ = 8%. The

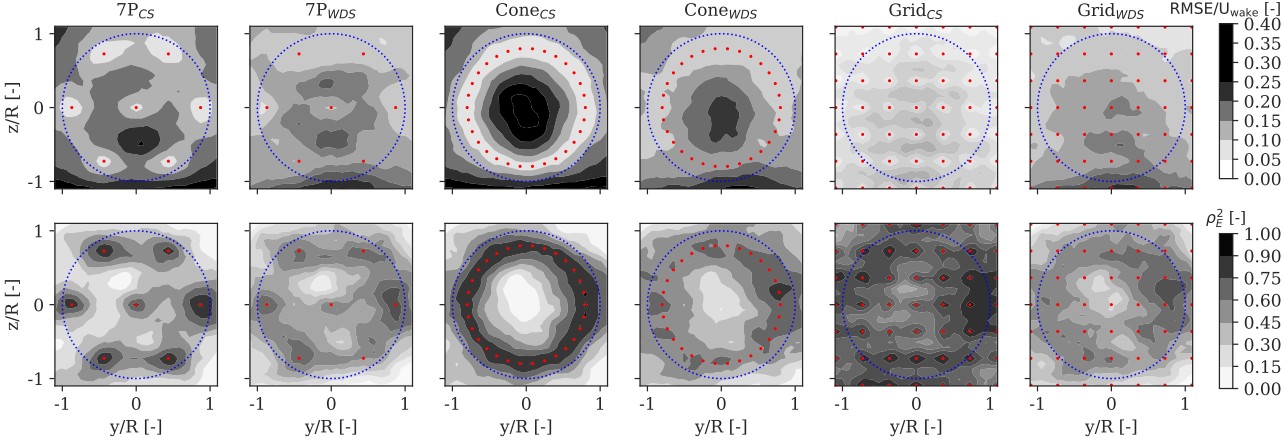

**Figure 5.** Spatial distribution of the error inherent to the *CS*- and *WDS*-reconstructed fields for selected scanning configurations. The top row refers to the RMSE normalized over the *target* velocity at each grid point. The bottom row refers to the explained variance ratio. The red markers identify the centers of the lidar beam sampling volumes. The wind turbine rotor is shown in blue.

time-series of the virtual lidar measurements is also shown. We find that both field reconstruction approaches can predict the reduced wind speed within the wake region and recover the details of the wind speed fluctuations of the *target* field. However, uncertainty is introduced due to the limited lidar sampling frequency, the probe volume length (here assumed to be 30 m), and the adopted field reconstructing techniques. The results in Fig. 6 demonstrate that incorporating lidar data directly in the reconstructed field (i.e., the *CS*-approach) leads to reproducing more accurate fields compared to the *WDS*-approach.

In addition, we compute the power spectral density (PSD) of the above analyzed time series of $u$-velocity fluctuations for a 10-min simulation and compare them in Fig. 7. We observe that the PSD of the reconstructed fields is comparable to that of the *target* for frequencies up to $\approx 1$ Hz, while the energy spectral content at higher frequencies is attenuated. According to Larsen et al. (2008), the dominant frequency of the wake meandering is defined as $f_{cut,off} = U_{amb}/(2D) = 0.016$ Hz ($\sim 62$ s period) for $U_{amb} = 6$ m/s. As the lidar completes a full-scan in 2 s, the large-scale wake meandering dynamics are well-captured. Further, as the wake meandering is the main source of wake added turbulence (i.e., $u$-component variance), the energy spectral content in the low-frequency range is recovered, as shown in Fig. 7.

The enhanced turbulent energy content of the *target* field within the high-frequency range (> 1 Hz) originates from the small-scale wake added turbulence (Madsen et al., 2010; Chamorro et al., 2012). These effects are not fully recovered in the reconstructed fields, mainly due to the lidar probe volume and limited sampling frequency, and because the method fits the lidar measurements to a standard Mann turbulence field, without the small-scale wake-added turbulence being explicitly included.

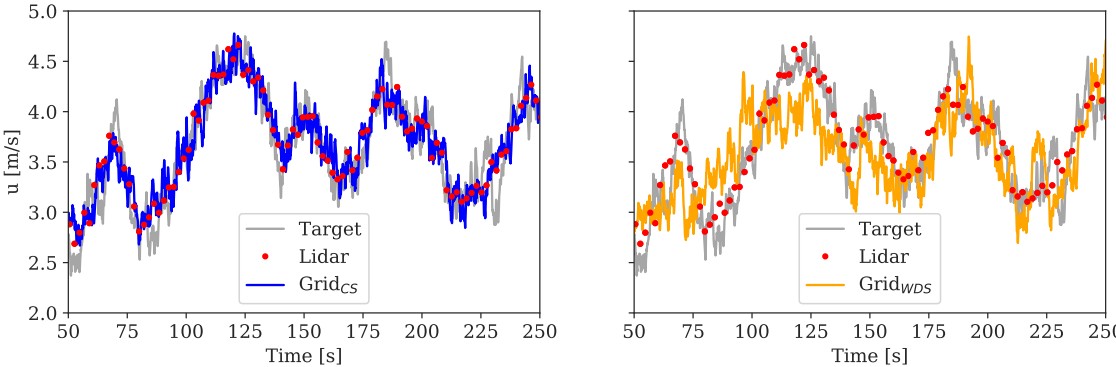

**Figure 6.** Comparison between the *target* $u$-velocity time-series at hub height (grey solid line) and the reconstructed field based on the *CS*-approach (left) and *WDS* (right) extracted at hub height. The lidar data are shown in red. The *target* simulations are run with $U_{amb} = 6$ m/s and $TI_{amb} = 8\%$.

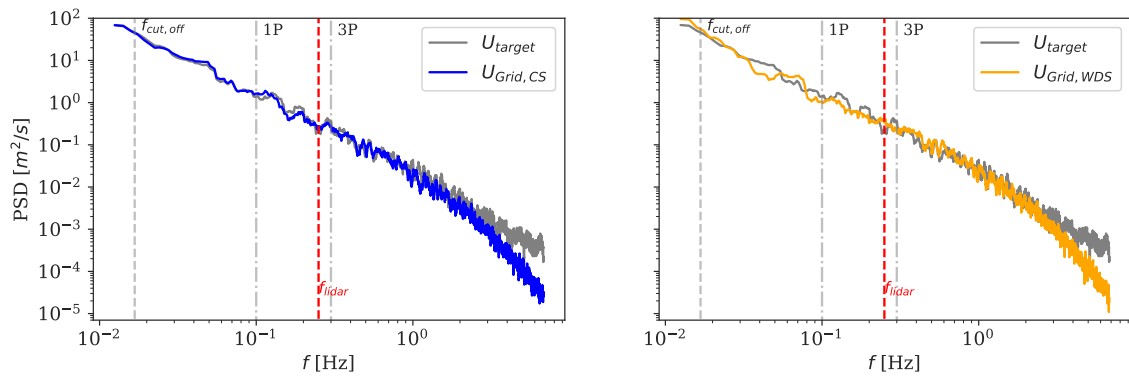

**Figure 7.** Comparisons of the power spectra density (PSD) of the *target* u-velocity component measured at hub height with predictions obtained by the *CS*-field (left) and the *WDS*-field (right). The dominant frequency of the wake meandering $f_{cut,off} \approx 0.016$ Hz, the rotational frequency of the rotor and its harmonics (1P $\approx 0.1$ Hz and 3P $\approx 0.3$ Hz), and the Nyquist frequency of the lidar ($\approx 0.25$ Hz) are shown (see text for more details).

### 4.2 Load validation

The DTU 10 MW reference wind turbine is used for the load validation analysis (Bak et al., 2013). The load simulations are carried out using the aeroelastic code HAWC2 (Larsen and Hansen, 2007) and inflow wind conditions measured from
an offshore site, as described in Sect. 4.2.1. Note that we run the analysis based on offshore wind conditions, which are

characterized by low turbulence; thus, wake effects are more prominent. This work evaluates the load prediction accuracy at the main wind turbine structures, such as blades, shaft, and tower. Therefore, we neglect the modelling of the offshore substructures and foundations, and we use the onshore model of the DTU 10 MW.

Following the load validation procedure illustrated in Fig. 1, we quantify the uncertainty in power and load predictions resulting from the *baseline*, *CS* and *WDS* simulations against results obtained with the *target* fields. The *CS* and *WDS* simulations are evaluated for the selected lidar configurations of Fig. 4, i.e., the 4P, 7P, Cone, SL, Grid and Grid* patterns with the parameters provided in Table 1. Two uncertainty indicators are defined to verify the load validation criteria *I* and *II* of Sect. 2:

- Bias: $\Delta_R = E(\tilde{y})/E(\hat{y})$,

- Uncertainty: $X_R = \sqrt{\langle (\tilde{y}/\hat{y} - E(\tilde{y})/E(\hat{y}))^2 \rangle}$,

where the symbol $E(.)$ denotes the mean value and $\langle . \rangle$ the ensemble average, $\hat{y}$ is the quantity of interest (i.e., power or load statistics) derived from the *target* simulations, and $\tilde{y}$ corresponds to that produced by the reconstructed fields. We evaluate $\Delta_R$ and $X_R$ on the resulting 10-min power and load statistics and provide results in Sect. 4.2.2 for the *CS*-fields, and in Sect. 4.2.3 for the *WDS*-fields.

The analyzed wind turbine responses include mean power production levels ($\mathrm{Power_{mean}}$), and fatigue loads. We use the rainflow counting algorithm to compute the 1-Hz damage equivalent fatigue loads with a Wöhler exponent of m = 12 for blades and m = 4 for steel structures as tower and shaft. Thus, we compute fatigue loads at the blade root flapwise and edgewise moments $\mathrm{MxBR_{DEL}}, \mathrm{MyBR_{DEL}}$, tower-bottom fore-aft and side-side $\mathrm{MxTB_{DEL}}, \mathrm{MyTB_{DEL}}$, the torsional loads at the tower top (also referred to as yaw moment) $\mathrm{MzTT_{DEL}}$ and torsional loads at the drivetrain $\mathrm{MzSh_{DEL}}$.

Furthermore, we quantify the accuracy of the reconstructed wake fields based on estimates of the rotor-effective wind speed ($U_{eff}$), defined as the weighted sum of the $u$-velocity measured across the rotor area, the explained variance ratio $\rho_E^2$, and the $u$-velocity variance $\sigma_u^2$ computed from the reconstructed turbulence fields. Finally, a load time-series and spectral analysis is conducted in Sects. 4.2.4 and 4.2.5.

### 4.2.1 Site conditions

Load simulations are carried out using site-specific observations collected from the FINO1 meteorological mast installed at the German offshore wind farm Alpha Ventus. The wind farm is situated in the North Sea and about 45 km north of the island of Borkum (Kretschmer et al., 2019). Data were collected over a period of three years from 2011 to 2014, and their details can be found in Kretschmer et al. (2019).

In the present work, we only use wind speeds and turbulence intensities measured under near-neutral conditions from a 90-m sonic anemometer installed at the mast. We extract 10-min average turbulence values binned for wind speeds ranging between 6 and 22 m/s; using wind speed bins with 2 m/s bin width we obtain nine bins with turbulence intensities of 8, 7, 7, 6, 6, 6, 6, 5, and 5%, respectively. These are the statistics of the ambient wind field that we use as inputs for the load validation analysis.

For each 10-min sample of the inflow wind, we use 18 turbulence field realizations (the IEC 61400-1 recommends at least 6 realizations), leading to 162 aeroelastic simulations for each analyzed scanning configuration. Simulations with ambient wind

speeds below 6 m/s are disregarded, as the wind speed approaching the rotor drops below the turbine's cut-in threshold due to
wake deficit effects, and the turbine shuts down.

Note that the recorded turbulence estimates at Alpha Ventus are considerably lower (approximately a factor of 3) than values
recommended by the low turbulence IEC-class C. Here, we perform the load validation analysis on more realistic turbulence
estimates characterizing offshore sites, since IEC-class C conditions would significantly attenuate the wake-induced effects, as
higher ambient turbulence leads to a faster recovery of the wake deficit.

We use standard IEC-recommended turbulence parameters for the Mann model (i.e., $L = 29.4$ m and $\Gamma = 3.9$ (IEC, 2019)),
whereas $\alpha_k \epsilon^{2/3}$ is tuned to obtain the *target* ambient turbulence levels of each simulation. The inflow is described by a power-
law with fixed shear exponent of $\alpha = 0.2$, as recommended in the IEC standard. The spacing between the analyzed and upstream
turbines is fixed at 5 D.

The *target* wake field characteristics as function of the ambient wind speed, which result from the 162 simulations, are shown
in Fig. 8. The wake considerably reduces the inflow wind speed approaching the rotor (i.e., $U_{eff}$) by $\approx 35\%$, compared to the
ambient wind speed (see Fig. 8-(**a**)). This effect decreases for higher winds ($> 14$ m/s) due to the low thrust coefficients of the
turbine. However, the wake deficit does not fully recover at high wind speeds ($U_{eff}/U_{amb} \approx 0.93$), as we simulate relatively
low ambient turbulence levels, the spacing between the turbines is short (i.e., 5 D), and the thrust coefficient of the turbine is
nearly constant at high wind speeds. Further, the contribution of the wind shear to the ratio $U_{eff}/U_{amb}$ accounts for up to 3%
in free-stream conditions, i.e., $U_{eff}/U_{amb} \approx 0.97$.

Turbulence levels within the wake region are nearly doubled at low wind speeds compared to the ambient conditions, as
shown in Fig. 8-(**b**). Further, the wake meandering amplitudes, here computed as the standard deviation of the wake center
displacements in the transverse directions normalized with the rotor diameter ($\sigma_{\mu_y}/D$ and $\sigma_{\mu_z}/D$), are also shown in Fig. 8-
(**c**),(**d**). As expected, larger wake displacements occur in the lateral than in the vertical directions (Keck et al., 2014; Machefaux
et al., 2016).

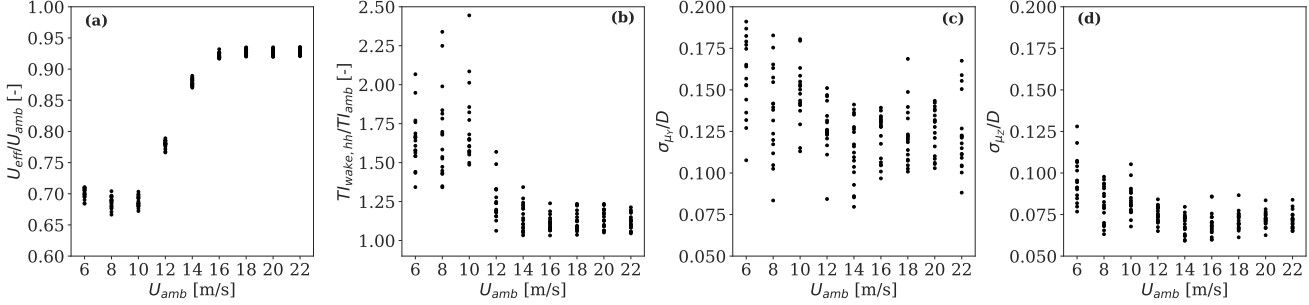

**Figure 8.** Scatter plots of the 10-min wake field characteristics resulting from the 162 simulations used as *target* in the load analysis. The
parameter $TI_{wake,hh}$ refers to the turbulence intensity measured at hub height in the wake; $\sigma_{\mu_y}/D$ is a measure of the amplitude of wake
meandering in the lateral direction and $\sigma_{\mu_z}/D$ refers to the vertical displacement of the wake.

### 4.2.2 Load uncertainty of constrained Gaussian wake field simulations

The uncertainties ($\Delta_R$ and $X_R$) of load predictions obtained with the *CS*-fields as a function of the ambient wind speed are shown in Fig. 9. We find that the biases largely vary depending on the simulated scanning pattern and analyzed load sensor. First, we observe that the patterns with fewer 'points' (i.e., 4P, 7P and Cone) overestimate $U_{eff}$ by 2–10% (see Fig. 9a). This is because 1) these patterns scan an insufficient amount of positions within the inflow area to characterize the wake flow fully; 2) the autocorrelation structure of the unconstrained turbulence box is such that the spatial regions that are not scanned by the lidar are nearly uncorrelated with the locations of the imposed constraints, as also shown in Fig. 5. Thus, in the regions that are not scanned by the lidar, the reconstructed wind speed approaches the ambient wind speed values.

As a result, lower deficits are simulated, or equivalently higher rotor-effective wind speeds are predicted. Consequently, the power predictions are overestimated ($\Delta_R$ >10%), as seen for ambient wind speeds below 14 m/s in Fig. 9b. Patterns with high spatial resolution, as the SL, Grid, and Grid*, provide rotor-effective wind speed and power production estimates in good agreement with the *baseline*.

The statistics of $\rho_E^2$ in Fig. 9c indicate that increasing the amount of points scanned by the lidar (see SL, Grid and Grid*) leads to a more accurate reconstruction of the wake turbulence. The biases of both $\rho_E^2$ and $U_{eff}$ decrease for high wind speeds due to the attenuated wake-induced effects (see Fig. 8). The improved performance of the SL, Grid (and Grid*) is also confirmed by estimates of $\sigma_u^2$ in Fig. 9d, which show that the SL and Grid configurations can match the *target* variance with an accuracy up to 98%, compared to 40–60% accuracy inherent of the 4P, 7P, and the Cone configurations. Nevertheless, the observed biases of $U_{eff}$, $\rho_E^2$ and $\sigma_u^2$ reveal that the 4P, 7P, and Cone patterns lead to inaccurate wake field representations and do not satisfy the criteria of the load validation (see Criteria I in Sect. 2).

The results from simulations with the SL, Grid and Grid* patterns provide fatigue load statistics of $\mathrm{MxBR_{DEL}}$, $\mathrm{MxTB_{DEL}}$, $\mathrm{MzTT_{DEL}}$ and $\mathrm{MzSh_{DEL}}$ in good agreement with the results of the *baseline* (see Fig. 9e–h). However, the calculated biases indicate a consistent underprediction at all wind speeds. This gap is largely compensated when probe volume effects are neglected, as seen for the Grid* (green lines). Overall, the observed deviations in the load predictions are due to the uncertainty of lidar measurements (i.e., size of the probe volume, cross-contamination effects, limited sampling frequency) and the limited scanning coverage of the patterns.

Figure 10 shows the statistics of $\Delta_R$ and $X_R$ including all wind speeds. As expected, the *baseline* leads to $\Delta_R \sim 1$ for all the analyzed load sensors, which indicates that the adopted 18 turbulence seeds are sufficient for the load statistics to converge. The large biases from simulations with the 4P, 7P, and Cone patterns ($\Delta_R \sim 0.87$–1.37) follow from the inaccurate wind field reconstruction discussed above. The load predictions with the SL and the Grid configurations provide biases closer to the *baseline*, although turbulence-driven load sensors are underpredicted by 2–7%. These deviations decrease as probe volume effects are neglected (i.e., $\Delta_R \sim 1\%$ for Grid* in Fig. 10-left).

The statistics of $X_R$ are shown in Fig. 10-right. The *baseline*'s $X_R$ is a direct measure of the statistical uncertainty intrinsic of the DWM model, which is due to the stochastic properties of the synthetic turbulence field and wake meandering. Thus, the turbine responses largely affected by wake-induced effects are identified by high $X_R$ values (see *baseline* in Fig. 10-right). The

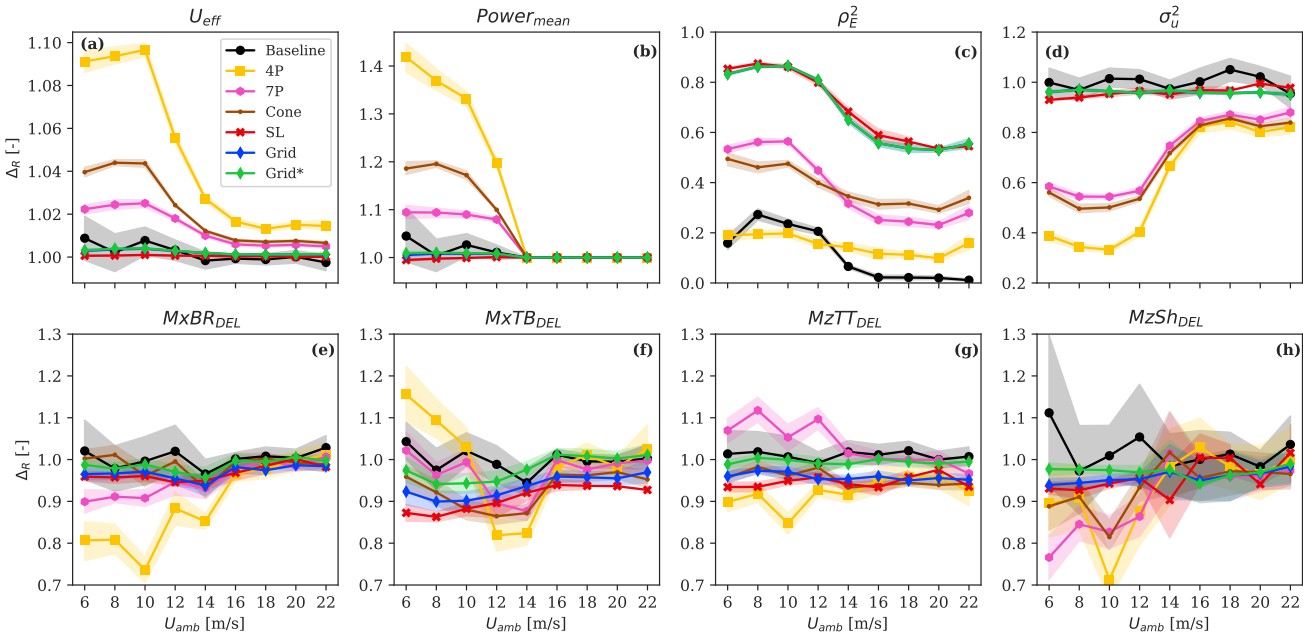

**Figure 9.** Comparison of bias $\Delta_R$ (solid line) and uncertainty $X_R$ (error band) of selected load sensors as function of the ambient wind speed. The uncertainty indicators are computed against the *target* observations, for each ambient wind speed (marker) that consists in 18 aeroelastic simulations with random turbulence seeds. The lidar-based results are derived from simulations with *CS*-fields.

power predictions and the majority of fatigue loads show a relatively high statistical uncertainty ($X_R \sim 0.05$–$0.09$), resulting in a large load scatter.

The $X_R$ values of MyTB and MzSh are significantly higher than other load sensors. The cause of the former is structural resonance occurring at low wind speeds for which the 3P frequency ($\approx 0.3$ Hz) excites the tower's natural frequency ($\approx 0.25$ Hz) (Bak et al., 2013). This effect originates from a design aspect of the DTU 10 MW turbine, and is amplified under wake conditions due to the induced unbalanced aerodynamic load distribution at the rotor. Nevertheless, structural resonance is independent of the wake-field reconstructing approach. The high $X_R$ values of MzSh originate from the intense controller activity to regulate the generator torque under high-variable inflow conditions. Significantly lower $X_R$ values characterize the *CS*-based load predictions compared to the statistics obtained with the *baseline*. $X_R$ values are reduced by a factor between 1.4–5 for the main wind-driven turbine responses such as $\text{Power}_{\text{mean}}$ and fatigue loads (i.e. $\text{MxBR}_{\text{DEL}}$, $\text{MxTB}_{\text{DEL}}$, $\text{MzTT}_{\text{DEL}}$ and $\text{MzSh}_{\text{DEL}}$). The *CS*-fields, reconstructed using scanning patterns with a sufficient amount of scanned positions and limited lidar probe volume, can satisfy both the load validation Criteria I and II in Sect. 2.

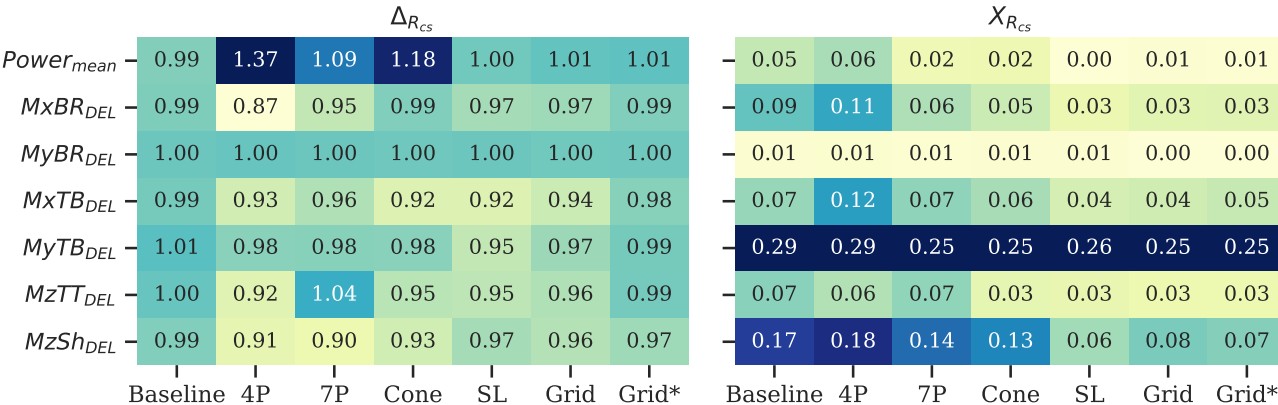

**Figure 10.** Uncertainty indicators of the load validation analysis based on the constrained field simulations (*CS*). Results are tabulated according to the load components and lidar scanning patterns. The colormap reflects the amplitude of the error, thus a dark blue identifies an overprediction while the ligth-green indicates an undeprediction. A perfect statistical prediction leads to $\Delta_R = 1$ and $X_R = 0$.

### 4.2.3 Load uncertainty of wake deficit superposition simulations

We present the results relative to the *WDS* simulations in the same fashion as for the *CS* in Sect. 4.2.2. Thus, we plot the load prediction uncertainty as a function of the ambient wind speeds in Fig. 11. The 4P, 7P and Cone patterns lead to improved biases of $U_{eff}$, and consequently $\text{Power}_\text{mean}$ (see Fig. 11a,b) compared to the results obtained with the *CS* fields (shown in Fig. 10). The $\text{Power}_\text{mean}$ predictions computed with the *WDS*-approach and the 7P pattern are comparable with the *baseline*, while the results using the *CS*-fields overpredicted it by $\approx 10\%$. Also, improved estimates of both $\rho_E^2$ and $\sigma_u^2$ are found in Fig. 11c,d, and for low wind speeds, which indicates a more accurate reconstruction of the wake turbulence by the *WDS*- than the *CS*-approach. In contrast, we find lower values of $\rho_E^2$ and $\sigma_u^2$ under higher compared to lower wind speeds because of the less pronounced wake deficits of the *target* fields, as also shown in Fig. 8-left.

These findings suggest that more details on the wake characteristics are better recovered by fitting a wake deficit function rather than incorporating lidar measurements directly into the turbulence boxes, when looking at patterns where the inflow is scanned at few positions. Overall, simulations with the 7P, SL, Grid, and Grid* patterns can produce power predictions comparable with the *baseline* (see Fig. 11b), whereas the 4P and Cone patterns lead to inaccurate predictions. Figure 11e – h shows that the fatigue loads obtained with the 7P, SL, Grid, and Grid* configurations are generally lower than those from the *baseline*.

We quantify the statistics of $\Delta_R$ and $X_R$, including all the wind speeds using *WDS*-simulated fields, and present the results in Fig. 12. As discussed above, the 4P and Cone patterns overpredict the rotor-effective wind speed and underpredict the wake turbulence; these effects counteract each other leading to fictitious lack of biases of fatigue loads. Similar conclusions can be made for the 7P configuration, although it provides reliable power estimates.

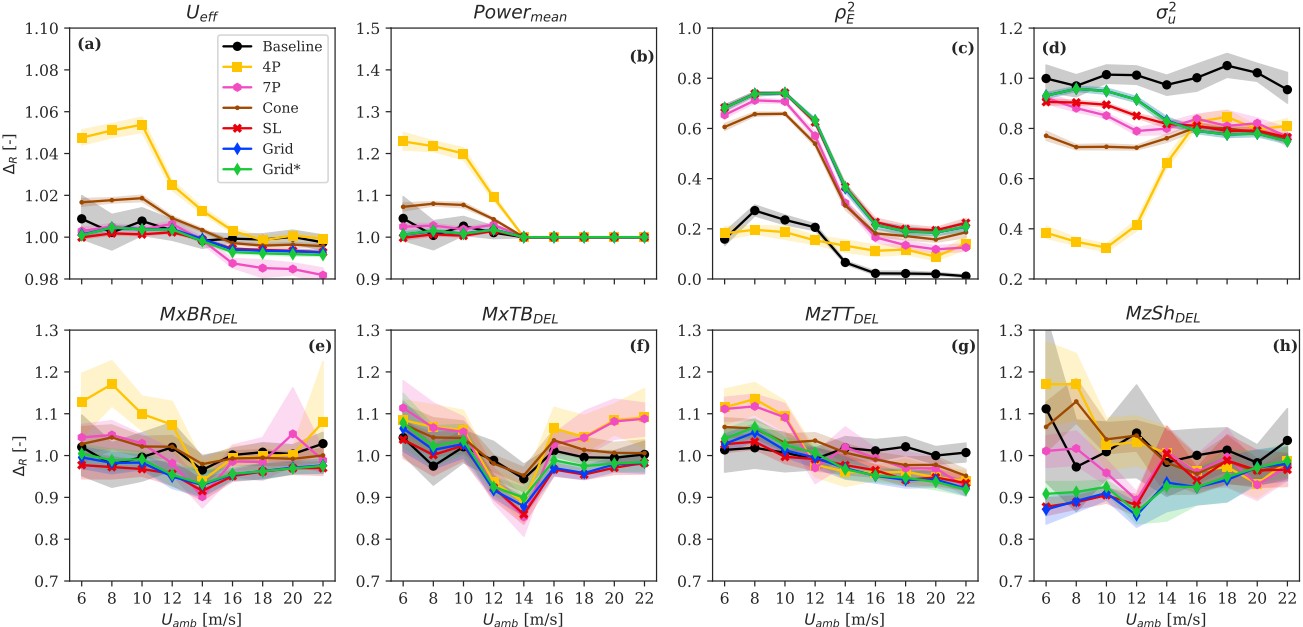

**Figure 11.** Comparison of bias $\Delta_R$ (solid line) and uncertainty $X_R$ (error band) of selected load sensors as function of the ambient wind speed. The uncertainty indicators are computed against the *target* observations, for each ambient wind speed (marker) that consists in 18 aeroelastic simulations with random turbulence seeds. The lidar-based results are derived from simulations with *WDS*-fields.

As seen for the *CS*-results, the SL, Grid, and Grid\* configurations provide biases in good agreement with the *baseline*, although fatigue loads are underpredicted up to 6%. By neglecting volume-averaging effects (i.e., Grid\*), only a marginal improvement of the biases is achieved. Simulations with the *WDS*-fields can reduce the statistical uncertainty of $\mathrm{Power_{mean}}$ by a factor of 5 and that of the main load components (i.e., $\mathrm{MxBR_{DEL}}$, $\mathrm{MxTB_{DEL}}$, $\mathrm{MzTT_{DEL}}$ and $\mathrm{MzSh_{DEL}}$) by a factor of
1.2–2 compared to the *baseline* (see $X_R$ in Fig. 12-right).

### 4.2.4 Time-series analysis of load predictions

In this section, we investigate the accuracy of lidar-reconstructed load time-series against *target* observations. An illustrative example is provided in Fig. 13, where the lidar-based power and loads time-series predictions are compared with the *target* simulations. As shown, both *CS*- and *WDS*-approaches recover to a large extent wake-induced effects and the instantaneous
events on the wind turbine responses, leading to load time-series that are highly correlated with the *target* observations. This finding explains the reductions of $X_R$ observed in Figs. 10 and 12.

In order to quantify the accuracy of the predicted load time-series, we evaluate the cross-correlations $\rho(\tilde{y}, \hat{y}) = \mathrm{cov}(\tilde{y}, \hat{y})/\sigma_{\tilde{y}}\sigma_{\hat{y}}$ between the lidar-based results ($\tilde{y}$) and the *target* simulations ($\hat{y}$) ($\rho = 1$ means perfect correlation). We focus the analysis on the SL, Grid and Grid\* configurations, which provide the most promising results, as demonstrated in the previous sections.

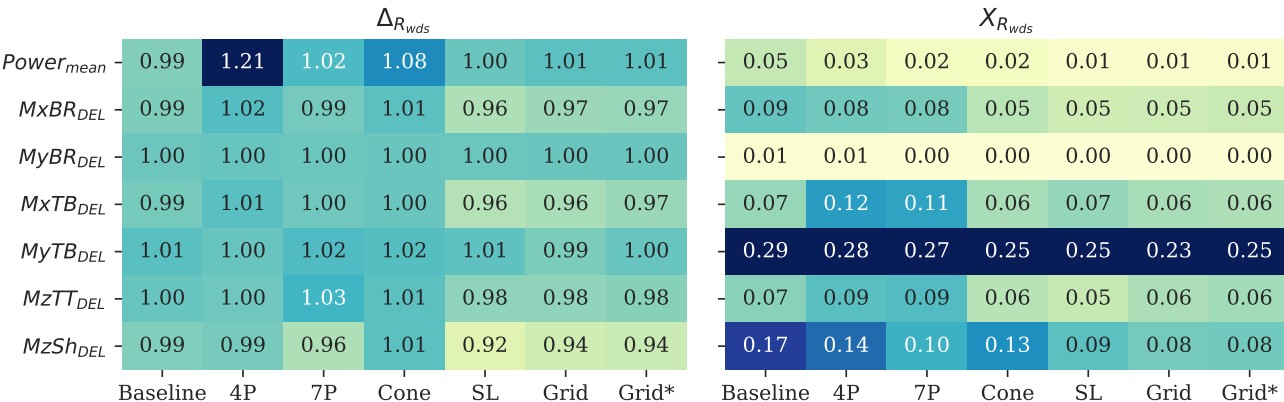

**Figure 12.** Uncertainty indicators of the load validation analysis based on the wake deficit superposition simulations (*WDS*). Results are tabulated according to the load components and lidar scanning patterns. The color-map reflects the amplitude of the error, thus a dark blue identifies an overprediction while the ligth-green indicates an undeprediction. A perfect statistical prediction leads to $\Delta_R = 1$ and $X_R = 0$.

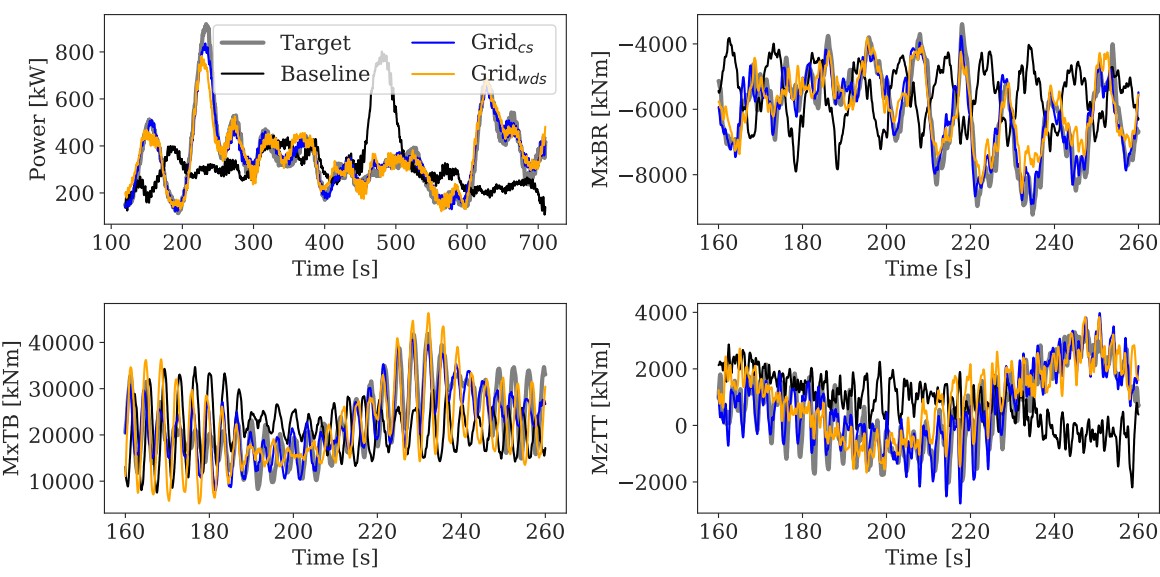

**Figure 13.** Comparison of predicted load time-series based on aeroelastic simulations carried out with the *target*, *baseline*, *CS*- and *WDS*-reconstructed fields. The lidar-based fields are reconstructed using the Grid pattern.

We compute $\rho$ for all the 162 simulations and for each load component, and provide average estimates in Fig. 14. We find that both the *CS*- and *WDS*-predicted Power time-series reach a nearly perfect correlation with the actual *target* observations ($\rho$ = 0.96–0.99). Note that Power is a low frequency signal (see Fig. 13), which is marginally affected by the local turbulence fluc-

tuations. A high correlation value is also obtained for MxBR ($\rho = 0.89$–$0.98$), and for the tower top and shaft load components ($\rho = 0.60$–$0.90$).

The correlation relative to MxTB drops to $\approx 0.33$ with the *WDS*-simulations, while higher values are achieved by the *CS* results. It should be noted that the structural resonance occurring at low wind speeds, which excites the tower can potentially affect the correlation results (Bak et al., 2013). Figure 13 shows that the MxTB time-series presents a nearly periodic signal, where the wind turbulence imprint is marginal. Overall, the accuracy of lidar-reconstructed load time-series show a significantly higher degree of correlation with the *target* observations, compared to that achieved by the *baseline*. Furthermore, the *CS*-approach can predict more accurately the observed load fluctuations compared to the *WDS*-approach.

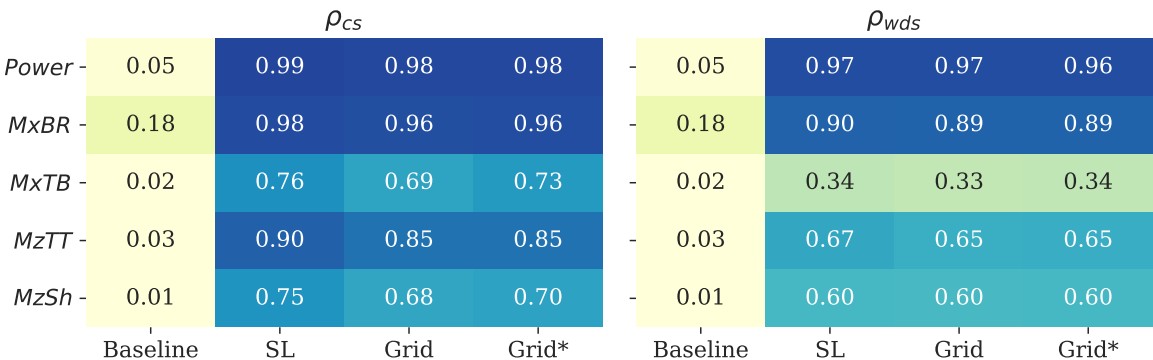

**Figure 14.** Average cross-correlation coefficient ($\rho$) computed between the reconstructed and *target* load times-series from all the 162 simulations. Results from the *CS*-fields are shown in the left panel, and those from the *WDS*-fields in the right.

### 4.2.5 Spectral coherence analysis of load predictions

We conduct a spectral analysis on the time-series of MxBR, MxTB, and MzTT, which are highly correlated with the wake meandering (Muller et al., 2015; Moens et al., 2019; Ning and Wan, 2019) and are primarily affected by wake turbulence. The PSD analysis is provided in Appendix A and shows that neither of the wake field reconstruction methods shift the energy content among frequencies nor introduce instabilities (i.e., artificial artifacts).

The spectral coherence analysis provides more insight on the accuracy of reconstructed blade and tower loads. Here, we compute the coherence as $\gamma^2 = |S(\tilde{y}, \hat{y})(f)|^2/(S(\tilde{y})(f)S(\hat{y})(f))$, where $S(\tilde{y})$ and $S(\hat{y})$ are the auto-spectra of the *CS* (or *WDS*) and *target* load estimates, and $S(\tilde{y}, \hat{y})$ is their cross-spectrum. We compare the coherence resulting from the load time-series produced by either *CS* and *WDS* simulations with the *target* observations for $U_{amb} = 6$ m/s and $TI_{amb} = 8\%$ (see Fig. 15).

It is observed that both field reconstruction techniques lead to high coherence in the proximity of the principal load frequencies, such as the rotational (1P for the blade, and 3P for the tower, see Fig. A1 for more details), the natural frequency of the tower ($\approx 0.25$ Hz, which is close to the 3P at 6 m/s), and the dominant wake meandering frequency ($\approx 0.016$ Hz). In

general, the coherence from the *CS* simulations is non-zero at frequencies up to 0.7 Hz (6P) and is higher than that from *WDS*
simulations. This confirms that higher frequency fluctuations can be reconstructed more accurately using the *CS* approach.

By increasing the scanning pattern's temporal resolution and the number of scanned points, and neglecting volume-averaging effects, the *CS*-approach could potentially reconstruct the whole spectrum of the loads. With the *WDS*-approach, we can only reconstruct turbulence structures corresponding to the size of the wake deficit. Finally, given the limitation of the reconstruction techniques to recover small-scale turbulence structures, as discussed in Fig. 7, the accuracy of tower loads, which are driven by
high-frequency fluctuations (see Fig. A1), is lower compared to that of the blades. This can partly explain the larger deviations of $\Delta_R$, $X_R$ and $\rho$, inherent of $\mathrm{MxTB_{DEL}}$ and relative to $\mathrm{MxBR_{DEL}}$, observed in Figs. 10, 12 and 14, as well as explaining why $\Delta_R$ for $\mathrm{MxTB_{DEL}}$ improves the most when the probe volume size is neglected, as seen in Fig. 10.

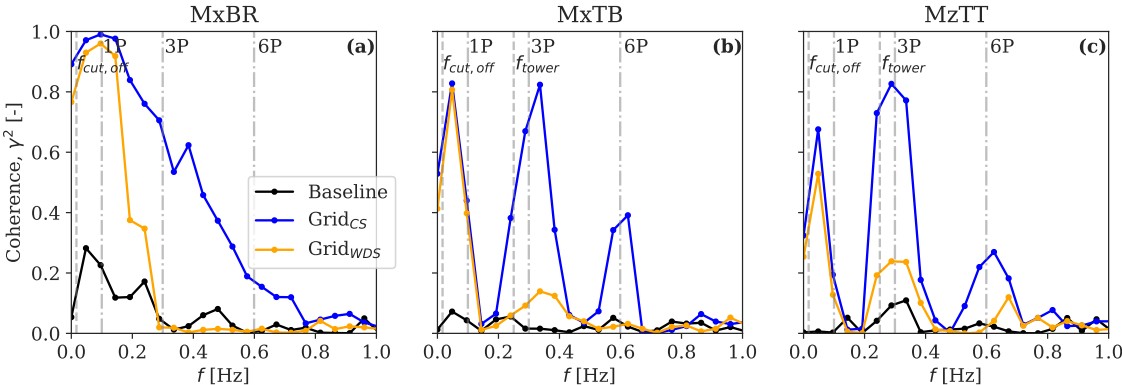

**Figure 15.** Spectral coherence analysis between the lidar-based load predictions and the *target* simulations for **(a)** the blade root flapwise bending moment $\mathrm{MxBR}$, **(b)** tower bottom fore-aft bending moment $\mathrm{MxTB}$, and **(c)** yaw moment $\mathrm{MzTT}$. The *target* simulations are run for $U_{amb}$ = 6 m/s and $TI_{amb}$ = 8%. The *baseline*'s results are also shown in solid black line, together with the principal operational frequencies of the wind turbine (1P $\approx$ 0.1 Hz, 3P and 6P) in dash-dot grey lines, the dominant frequency of the wake meandering $f_{cut,off} \approx 0.016$ Hz, and the natural frequency of the tower $f_{tower} \approx 0.25$ Hz in dashed grey lines.

## 4.3  Sensitivity analysis

The load validation of Sect. 4.2 is carried out using statistics collected under near-neutral conditions at Alpha Ventus, i.e.,
low atmospheric turbulence. Nevertheless, atmospheric turbulence conditions have a strong impact on the wake development (Kumer et al., 2017; Zhan et al., 2020), and wind turbine loads (Sathe et al., 2013; Kretschmer et al., 2018). Further, the lidar measuring characteristics can impact the accuracy of reconstructed fields (Lundquist et al., 2015), thus that of load predictions. In the next subsections, we investigate the sensitivity of atmospheric turbulence conditions as well as selected lidar specifications on the accuracy of lidar-based load predictions using the Grid pattern as an example.

 **4.3.1 Effect of atmospheric turbulence conditions on load prediction accuracy**

Figure 16a shows the sensitivity of the lidar-based load predictions bias on $TI_{amb}$ within the range 4–20%. The high $TI_{amb}$ leads to faster recovery of the velocity deficit (Doubrawa et al., 2019), amplifies the wake meandering (Machefaux et al., 2016), and affects the accuracy of lidar-reconstructed fields (Pettas et al., 2020). This has a negligible effect on the accuracy of load predictions obtained with the *CS*-fields, while larger deviations are observed for the *WDS* results. This is partly due to 605 the limited scanned area by the lidar combined with the large wake displacements. The fitting procedure intrinsic of the *WDS* approach can lead to an inaccurate estimation of the wake shape parameters when the wake moves out of the scanned area (Trujillo et al., 2011).

We investigate the influence of the atmospheric turbulence length scale on the load prediction accuracy in Fig. 16b, by varying $L$ between 5 and 70 m. Earlier studies have shown the strong dependency of load statistics on the turbulence length 610 scales (Sathe et al., 2013; Dimitrov et al., 2017; Conti et al., 2020). Further, $L$ provides a measure of the scanning configuration resolution useful for performing constraints (Dimitrov and Natarajan, 2017).

The turbulence length scale affects the predicted statistics of the explained variance ratio of the *CS*-fields, which decreases from $\rho_E^2 \sim 0.8$ for $L = 29$ m to $\rho_E^2 \sim 0.6$ for $L = 5$ m (not shown). This indicates that when $L$ is low, the turbulence structure sizes fall below the sampling fidelity of the *CS* approach (note that the scanned points of the Grid configuration are separated 615 by 29 m as described in Sect. 3.3). Also, the turbulence structure sizes become small relative to the lidar probe volume, causing the lidar measurements to average out more of the turbulence. The *CS*-based load predictions' biases show a dependency on the turbulence length scales, while the *WDS*-fields are not significantly affected (see Fig. 16b).

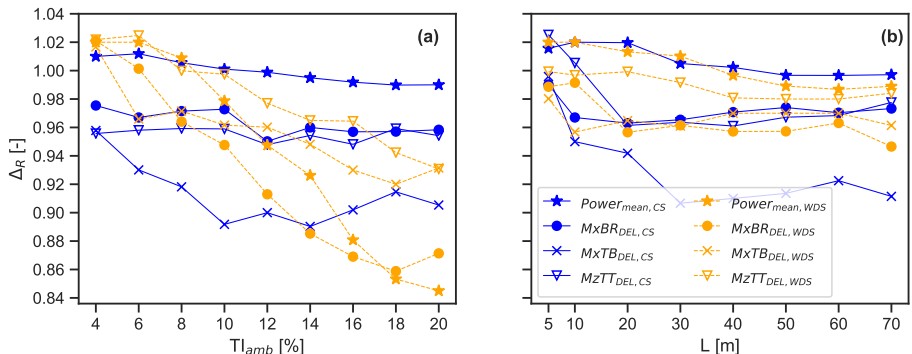

**Figure 16.** Influence of atmospheric turbulence conditions on the lidar-based load prediction accuracy, including: **(a)** the effect of ambient turbulence ($T_{amb}$) given $U_{amb} = 6$ m/s, **(b)** the effect of turbulence length scale, $L$, given $U_{amb} = 6$ m/s and $TI_{amb} = 8\%$. The bias $\Delta_R$ at each nominal value is computed from 18 simulated seeds. The Grid pattern is used for the analysis.

### 4.3.2 Effect of lidar probe volume and scanning period on load prediction accuracy

One of the main limitations of continuous-wave lidars is that the probe volume size increases proportionally with the square of the focal distance (Sathe and Mann, 2013). As the diameter of modern wind turbines has reached 150–200 m, measuring at farther distances upfront of the rotor becomes an issue due to the larger probe volumes. Hence, we investigate the sensitivity of the lidar probe volume on the load prediction accuracy in Fig. 17a, by varying the probe volume length between 0 to 210 m.

As shown, the magnitude of $\Delta_R$ decreases almost linearly with increasing probe volume lengths. Further, the probe volume effects are more pronounced for the *CS*-approach, which directly incorporates the low-pass filtered wind speed fluctuations into the reconstructed field.

Another limitation inherent of the pulsed lidar technology is the reduced sampling frequency compared to continuous-wave lidars (Peña et al., 2015). The lidar sampling frequency sensitivity on the load prediction accuracy is assessed by varying the scanning period, which is defined as the time to complete a full scan (1–30 s). For this particular analysis, the *target* simulations are run for $U_{amb}$ = 6 m/s and $TI_{amb}$ = 16%.

Although the scanning period does not play an important contribution to the load prediction accuracy, as shown in Fig. 17-b, this outcome is conditional to the dominant frequency of the wake meandering, which in turn decreases with larger rotors ($f_{cut,out} = U_{amb}/(2D)$), and the amount of scanned points in the inflow. The *CS*-results show that a bias lower than 2% in power predictions is found for scanning periods up to ≈20 s, which corresponds to one-third of the wake meandering dominant period.

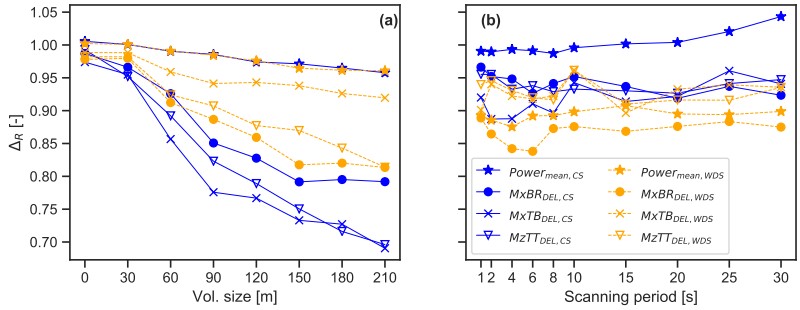

**Figure 17.** Influence of lidar scanning specifications on the lidar-based load prediction accuracy, including: **(a)** the effect of probe volume size given $U_{amb}$ = 6 m/s and $TI_{amb}$ = 8%, **(b)** the effect of the scanning period given $U_{amb}$ = 6 m/s and $TI_{amb}$ = 16%. The bias $\Delta_R$ at each nominal value is computed from 18 simulated seeds. The Grid pattern is used for the analysis.

## 5  Discussion

One of the main elements used in the study is to consider as *target* the wake flow fields generated by the DWM model. The DWM model is a simplified engineering wake model subjected to modelling uncertainties. Although the mean wind velocity and turbulence fields in the far wake region can deviate from high-fidelity simulations (e.g., computational fluid dynamics, CFD) or field data, the calibration of the DWM model coefficients can considerably improve the accuracy and provide wake fields in good agreement with lidar observations (Reinwardt et al., 2020) and CFD simulations (Keck et al., 2012, 2014, 2015). The modelling uncertainty originated from an inaccurate calibration of the DWM model is not expected to significantly alter this study's findings, as we demonstrate the robustness of the lidar-based approaches under a large variety of inflow wind and operational conditions.

The wake turbulence spectral properties are described, to the extent needed for the load analysis, by an isotropic Mann-generated turbulence field with a low length scale (Madsen et al., 2005). A more realistic modelling choice to accurately simulate the turbulence structures within the wake fields, which can also affect aeroelastic load simulations, is found in LES (Churchfield et al., 2015; Nebenführ and Davidson, 2017). Further, lidar-based wind field reconstruction techniques applied to LES fields have been recently developed (Bauweraerts and Meyers, 2020, 2021). Nevertheless, the computational burden of high-fidelity simulations, such as LES, would make the statistical load analysis of this work unfeasible.

Another limitation stems from the lidar simulator used in the study, which replaces full-field lidar measurements. Real lidar data taken upfront the rotor should be corrected for induction (Borraccino et al., 2017; Mann et al., 2018), blade blockage effects, and wind evolution (Bossanyi, 2013; de Mare and Mann, 2016). These effects are not simulated due to the modeling assumptions of the DWM model and should be further investigated, e.g., using LES fields. Despite the limitations mentioned above, the numerical framework developed within this work is useful to assess the influence of several uncertainty sources on power and load predictions and evaluate different lidar scanning strategies in an idealized yet fully controllable environment.

Characterizing the small-scale wake-added turbulence poses a challenge given the limitations of lidar's sampling frequency and probe volume size (Peña et al., 2017). The small-scale wake-added turbulence enhances the energy spectral content in the high-frequency range, 0.4–20 Hz (Madsen et al., 2010; Chamorro et al., 2012; Singh et al., 2014), and its contribution on the fatigue damage varies according to the load component and turbine operational strategy (Tibaldi et al., 2015). The *CS* algorithm could potentially reconstruct high-frequency wake-added turbulence. However, this would require sampling the wind field at a high temporal frequency and without probe volume effects. In contrast, the *WDS*-method cannot explicitly reconstruct the high-frequency wake-added turbulence.

Bergami and Gaunaa (2014) demonstrated that the most serious fatigue damage on the blades occurs at frequencies around 1P (0.1–0.16 Hz for the DTU 10 MW), whereas structures as tower top (nacelle) and tower bottom are mainly affected by the tower eigenfrequency ($\approx 0.25$ Hz) and the 3P frequency (0.3–0.48 Hz). As the PSD of tower loads exhibits large energy spectral content at high frequencies (see Fig. A1), the accuracy of tower load predictions decreases compared to that achieved by blade loads, as found in Sects. 4.2.4 and 4.2.5.

We demonstrate that a high number of lidar-scanned positions of the inflow is required to ensure an acceptable level of accuracy in the reconstructed wake fields. The results reveal that the current commercially available nacelle-mounted lidars (e.g., the 4P, 7P, and Cone patterns) will not provide sufficient information to reconstruct the wake fields accurately for the load assessments. In contrast, the scanning requirements are fulfilled by the SpinnerLidar and any arbitrary lidar that can potentially scan a greater region of the rotor, e.g., a Grid-like configuration. Although we do not optimize the scanning strategies, it is inferred that the required number of positions scanned by the lidar depends on the size of the turbulence structures in the wake field.

Incorporating a sufficient number of lidar measurements directly in the turbulence fields leads to more accurate load predictions than assuming a wake deficit's generic shape function. The *CS* algorithm can also be extended to reconstruct the $v$- and $w$-turbulence fluctuations (Dimitrov and Natarajan, 2017). Additionally, the *CS*-method finds direct application for reconstructing more complex flow fields occurring in wind farms, e.g., multiple wakes.

On the other hand, the accuracy of the *WDS*-predicted loads is conditional to the selected shape function's goodness to represent velocity deficits. The wake deficit can deviate from a Gaussian shape as the atmosphere becomes more unstable (Ning and Wan, 2019), it exhibits a double-peak shape in the near-wake region (Keck et al., 2014), and a more complex geometry in a multiple wake scenario. Overall, reproducing the actual observed wake meandering path in the wake field simulations can potentially reduce the statistical uncertainty of power and load predictions.

The fitting procedure of the *WDS*-approach is relatively fast and can provide real-time spatial and temporal characteristics of the wake flow field, which are useful for power and load predictions, wind farm monitoring, and control strategies. The computational cost of the *CS* algorithm considerably increases with the number of constraints simulated and the dimension of the turbulence boxes. For reference, a single wind field with 27900 constraints (i.e., using the SL configuration) and a turbulence box with a grid size of $8192 \times 32 \times 32$ points currently require one and a half-hour of simulation time on a single CPU.

## 6   Conclusions

This study proposed two alternative wind turbine load validation procedures under wake conditions that reconstruct synthetic wake fields from time-series of lidar retrievals. The first approach consisted of incorporating nacelle lidar measurements of the wake as constraints into random Mann turbulence field realizations. The second approach relied on the superposition of lidar-fitted bivariate Gaussian wake deficit time-series into the Mann turbulence fields.

The two approaches were numerically evaluated, adopting a tailored-designed framework that uses a virtual lidar simulator to scan three-dimensional wake fields simulated by the DWM model (i.e., the *target* fields).

We demonstrated that lidar-reconstructed wake fields recovered the main wake flow features affecting wind turbine power and load predictions, such as the spatial distribution of the velocity deficit and its meandering dynamics. However, the accuracy of power and load estimates was highly conditional on the amount of scanned points by the lidar, the probe volume size, and the ambient turbulence intensity that in turn affected the wake evolution.

The load validation analysis showed that the current commercially available nacelle-mounted lidars would not provide sufficient spatial resolution to characterize wakes for power and load assessments, whereas research lidars, e.g., the SpinnerLidar and the Grid-like configuration, fulfilled these requirements.

Provided that a sufficient number of wind measurements were taken upwind of the rotor (e.g., using the SpinnerLidar or the Grid), incorporating them as constraints into turbulence fields was the most robust and accurate procedure for reconstructing wake fields and predicting power and loads. The lidar-reconstructed wake fields produced power and load time-series that were highly correlated with the *target* turbine responses; thus, reducing the statistical uncertainty (realization-to-realization) by a factor 1.2–5 when compared to the traditional load validation procedure (i.e., using the DWM model).

Although unbiased power productions were predicted, the SpinnerLidar- and Grid-based reconstructed wake fields under-predicted fatigue load estimates by 1–8% depending on the load component and the size of the probe volume. The biases in fatigue load predictions were further reduced to less than 2% when neglecting probe volume effects.

Further investigations should evaluate the effects of rotor induction and turbulence evolution on the accuracy of lidar-reconstructed wake fields. Besides, the proposed wake field reconstruction techniques should be validated using full-field data collected in operating wind farms.

*Code availability.* The ViConDAR framework can be accessed at https://github.com/SWE-UniStuttgart/ViConDAR, while the constrained simulation framework at https://gitlab.windenergy.dtu.dk/nkdi/constrained-simulation.

## Appendix A: Power Spectral Density (PSD) of load predictions

Figure A1 shows a comparison of the PSD of $MxBR$, $MxTB$, and $MzTT$ between the lidar-reconstructed and *target* simulations for $U_{amb} = 6$ m/s and $TI_{amb} = 8\%$. Figure A1a displays the PSD of $MxBR$, where the first three peaks correspond to the subsequent rotor harmonics (1P, 2P and 3P). The highest observed peak is at 1P ($\sim 0.1$ Hz), which indicates that the greatest load cycle amplitude is due to asymmetric blade loading condition. This effect is amplified by the inhomogeneous wake field approaching the rotor.

Compared to the rotating blades, the PSD of the tower loads $MxTB$ and $MzTT$ exhibits the largest energy content at higher frequencies up to 3P ($\sim 0.3$ Hz). Further, the natural frequency of the tower (0.25 Hz) corresponds nearly to the 3P frequency at 6 m/s. This explains the very high peak seen for the $MxTB$. Overall, the PSD produced by the simulations with lidar-reconstructed fields ($CS$ and $WDS$) shows good agreement with that of the $target$ simulations, meaning that the energy content is not being shifted between frequencies. However, it is observed that the energy content at high frequency ($> 1$ Hz), induced by the wake-added turbulence, is not fully recovered due to the lidar probe volume and limited sampling frequency.

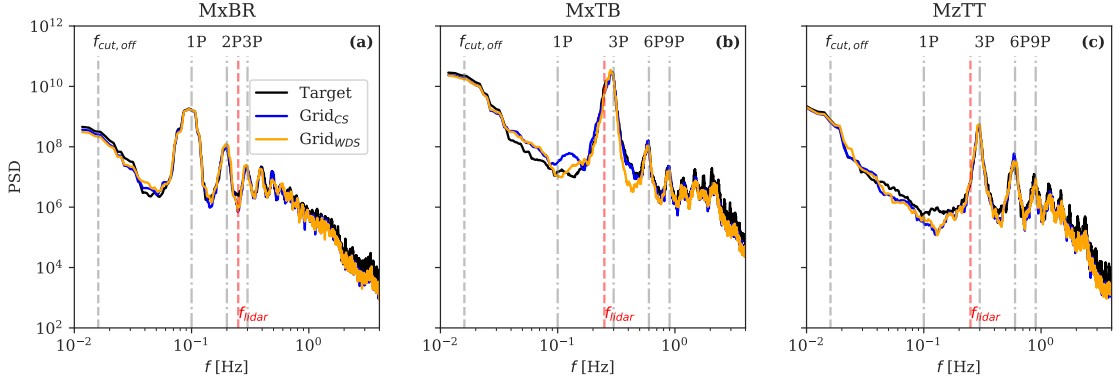

**Figure A1.** Power spectral density (PSD) of **(a)** the blade root flapwise bending moment MxBR, **(b)** tower bottom fore-aft bending moment MxTB, and **(c)** yaw moment MzTT. The dominant frequency of the wake meandering $f_{cut,off}$ = 0.016 Hz, the Nyquist frequency of the lidar $f_{lidar}$ = 0.25 Hz (corresponding to the scanning period), and the main rotational frequencies 1P, 2P, 3P, 6P and 9P are shown. The $target$ simulations are run at 6 m/s with $TI_{amb}$ = 8%.

*Author contributions.* DC, VP, ND, and AP participated in the conception and design of the work. VP developed the ViConDAR framework. ND developed the constrained simulation framework. DC developed the numerical framework that gets ViConDAR to scan wake-generated fields and use virtual lidar measurements as inputs to reconstruct wake field via *CS-* and *WDS*-approaches. DC also carried out aeroelastic simulations, and wrote the draft manuscript. VP, ND and AP provided key elements of the programming code, supported the overall analysis, and critically revised the manuscript.

*Competing interests.* The authors declare that they have no conflict of interest.

*Acknowledgements.* The work conducted by Vasilis Pettas is funded by the German Federal Ministry for Economic Affairs and Energy (BMWi) in the framework of the national joint research project RAVE - OWP Control (ref, 0324131B).

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
