# Peer review of "Wind turbine load validation in wakes using wind field reconstruction techniques and nacelle lidar wind retrievals"

_Wind Energy Science, 2020_

## Referee Comment (RC1) · Anonymous Referee #1 · 3 Nov 2020

This paper presents methods for performing load validation by reconstructing the wind field upstream of a waked wind turbine using nacelle lidar measurements. The load validation methods are simulated using the dynamic wake meandering (DWM) model and aeroelastic simulations, and compared to the performance of the standard IEC-recommended DWM method for load validation. The paper is a nice extension of previous work by the authors "Aeroelastic load validation in wake conditions using nacelle-mounted lidar measurements," where the authors use wind parameters based on lidar measurements in wake conditions to evaluate the accuracy of load validation as part of a field experiment.

[Figure]

The paper is well written and clearly organized. Furthermore, the topic is relevant given the interest in using nacelle lidars for applications such as power performance and load validation in the wind industry. By investigating the proposed load validation methods in a controlled simulation environment, the authors are able to isolate the impact of the wind field reconstruction methodology, without worrying about aeroelastic model uncertainties. Although there are no major issues with the paper, there are several smaller comments that I believe should be addressed by the authors.

First, more motivation for the proposed lidar-based load validation methods should be presented. For example, if the goal is to achieve load prediction biases that are the same as the baseline method but with lower statistical uncertainty, how will this improve the wind turbine design process? And can you discuss current problems with the IEC-recommended approaches for load validation in wake conditions?

Comments:

1. Title: Instead of "using field reconstruction techniques," which is somewhat vague, consider "using wind field reconstruction techniques"

2. Pg. 2, ln. 34: "the 10-min statistical properties (mean and variance) of the simulated ambient and operational conditions are set to match the measured ambient wind statistics": This doesn't quite make sense. How can the simulated "operational" conditions be set to match measured "ambient" conditions. Wouldn't you only need to match the ambient conditions?

3. Pg. 2, ln 45: "...which increases the amount of validation data." This could use a little more explanation (contrast this to a fixed met tower where only a small sector is valid).

4. Pg. 2, ln. 49: "The recent work of Conti et al. (2020) demonstrated that lidar-based load validation procedure in wakes should account for a model of the wake deficit and its dynamics." Since this paper builds on the work of Conti et al. 2020, please discuss

this work in a little more detail, especially why it was concluded that lidar-based load validation in wakes should include a wake model.

5. Pg. 3, ln. 83: "through a field reconstruction technique": "wind field?" Or "wake field?"

6. Pg. 4, ln. 98: "large number of simulations" How many?

7. Pg. 4, ln. 101: "The mean bias of load predictions… is of the same order of that obtained with the baseline". Please be more specific about how close the lidar-based simulations should be to the baseline. "Of the same order" is a subjective criteria and makes it hard to tell if the new methods are successful.

8. Pg. 4, ln. 111: "the IEC recommends, i.e., the Mann uniform shear spectral tensor model…" This is one model that is recommended. There is also the Kaimal spectral model, etc.

9. Eq. 1: the symbol "i" is used twice, for the spatial location as well as to indicate imaginary numbers. Can you choose unique symbols?

10. Eq. 2: Should the bold "k" argument on the left hand side be "k_1"?

11. Section 3.2: Can you explain more about the tools you are using to implement DWM? In other words, is DWM a software tool that you are using (if so, a reference would be appreciated)? Or is it a model described in the literature that you are implementing yourselves?

12. Fig. 1: What wind speed is used for the middle plot?

13. Pg. 6, ln. 157: "The latter increases the uncertainty of the procedure." It is unclear what "the latter" refers to here.

14. Pg. 6, ln. 166: "while the spatial resolution in the longitudinal axis depends on the simulated wind speed." Then what is the temporal resolution of the wind field?

15. Pg. 7, ln. 171: "continuous-wake" -> "continuous-wave"

16. Pg. 7, ln. 181: Can you provide a reference for the 4-beam Leosphere lidar?

17. Pg. 8, ln. 185: "7-beam lidar can potentially increase the accuracy of reconstructed wind fields." Increase the accuracy compared to what?

18. Pg. 8, ln. 193-194: There is also a 4-beam Windar CW lidar, and the grid-configuration pattern is based on the SWE pulsed lidar. Can you explain why you classified these scan patterns as pulsed and CW, respectively? Furthermore, since you are only modeling a single measurement range, it is unclear how you model CW and pulsed lidars any differently in you simulations. Can you explain this further? Lastly, you are giving up additional measurement points (and therefore potentially wind field reconstruction accuracy) by only using a single range for the pulsed lidars. Why didn't you use multiple range gates?

19. Pg. 8, ln. 199: "A preview distance of 0.7 D is assumed." In addition to the lidar measurement accuracy arguments, there seems to be an interesting dilemma when measuring the wake deficits upstream of a turbine. On one hand, I imagine you would want to measure close to the turbine to capture the true wake velocity deficit at the rotor plane. On the other hand, measuring too close will introduce induction zone effects. Can you discuss how you approached this issue?

20. Pg. 8, ln. 204: "A probe volume with an extension of 30 m in the LOS direction is assumed" Can you provide some references for how you chose 30 m for pulsed and CW lidars? Furthermore, how is the probe volume extension defined? For example, the std. dev. of Gaussian weighting function?

21. Pg. 9, ln. 218: "obtained by simply scaling an isotropic turbulence field..." Can you clarify if the scaling depends on the radial location from the wake center, as shown in Fig. 1?

22. Pg. 9, ln. 221: How might the ambient wind conditions be measured in practice?

[Figure]

23. Pg. 10, ln. 226: What do you mean by "The u-velocity fluctuations are recovered from the target wake fields?"

24. Pg. 11, ln. 256: "By denoting. . . as the constrained turbulence field that incorporates lidar measurements. . ." It seems that in Eq. 9, uˆprime_CS,B,i, represents the turbulent fluctuations with the mean ambient wind profile removed. Do you first remove the mean ambient wind speeds from the lidar measurements before they are used to generate the constrained turbulence field?

25. Eq. 10: I'm confused about how K_def,lidar is defined. From Fig. 1, K_def is presented as a scaling factor applied to the ambient wind field (= 1, when wake losses are not present). But here, it appears to be defined as the normalized deficit (= 0, when wake losses are not present). Can you clarify this and make sure the definitions of K_def are consistent?

26. Eq. 10: Since the left hand side of this equation is being fit to the Gaussian function, they are not actually "equal." It would make more sense to present this equation as a minimization objective function (e.g., based on the difference between the measured deficit and the Gaussian model) Also, should U_amb(z) have the mean operator applied to it, like in Eq. 9?

27. Eqs. 10 and 11: The explanation of Eq. 11 is confusing. In your final method are you using the Gaussian fit from Eq. 10 as part of Eq. 11, or does Eq. 11 entirely replace Eq. 10? It would help to present both equations as minimization problems, so it's easy to see where the lidar measurements are being used and what exactly is being fit to the Gaussian profile.

28. Section 4.1: In addition to the analyses presented, a nice way to quantify the accuracy of the reconstructed wind fields could be to compare the RMSE of the rotor average wind speed u_eff as well as the best-fit linear horizontal and vertical shear coefficient time series between the target and reconstructed wind fields. These variables should play a large role in determining the turbine loads.

29. Pg. 12, ln. 308: "run at the downstream distance of 5 D" Please be more specific. The turbine of interest is located 5 D downstream of the upstream turbine?

30. Fig. 7: On the left plot showing U_eff/U_amb, can you explain why the ratio converges to $\sim$0.93 at high wind speeds? As wind speed increases, the turbine thrust should keep decreasing causing wake losses to continue to decrease, so I would expect the ratio to approach 1.

31. Pg. 18, ln. 440: "In addition, improved estimates of both rhoˆ2_E and sigmaˆ2_u are seen in Fig. 10c,d." There seem to be improvements at low wind speeds, but slightly worse performance at high wind speeds. Can you comment on this in the paper?

32. Figs. 10 and 11: I would suggest full captions.

33. Pg. 20, ln. 463: "focus the analysis on the SL, Grid, and Grid* configurations" These might be the most promising scan patterns, but also not the most likely, given currently available commercial lidar technology. It would be interesting to analyze the time series for one of the commercially-available lidar scenarios as well.

34. Pg. 21, ln. 469: "It should be noted that the structural resonance occurring at low wind speeds, which excites the tower can potentially affect the correlation results." Can you discuss why this resonance appears? Could it be removed by improving the controller tuning?

35. Pg. 22, ln. 481: Usually magnitude-squared coherence is written as gammaˆ2 = abs(S_x,y)ˆ2/(S_x*S_y). Therefore, I would expect your definition to be gamma = abs(S_x,y)/sqrt(S_x*S_y). Is this correct?

36. Fig. 14: On the left plot, why is the baseline coherence so high at low frequencies (above the noise floor)?

37. Pg. 25, ln. 538: As mentioned earlier, the "need for reducing the statistical load prediction uncertainty" in wake conditions could be motivated more clearly in the paper.

More discussion or references talking about the need for improved methods would strengthen the message of the paper.

38. Pg. 26, ln. 562: You say that the lidar-based predicted load statistics are comparable to the results from the baseline DWM method (Delta_R between 0.97 - 1.01). However, from Figs. 9 and 11, it seems more accurate to say that Delta_R is between 0.92/0.94 and 1.01. Is 0.94 still an acceptable difference?

39. Pg. 27, ln. 610: Similarly, the range of Delta_R with the lidar-based method is more like 0.94-1.01 instead of 0.97-1.01. When saying that this is comparable with the baseline method, please be more specific about what "comparable" means.

40. Pg. 27, ln. 615: In addition to these lidar parameters, the load prediction accuracy is sensitive to the turbulence intensity as well.

41. Pg. 28, ln. 628: "largest energy content at higher frequencies (> 3P $\sim$ 0.3 Hz)." From the plots, the largest energy content is at very low frequencies and right at 3P, but > 3P does not contain as much energy content.

---

## Referee Comment (RC2) · Anonymous Referee #2 · 5 Dec 2020

**Review of the manuscript wes-2020-104, entitled "Wind turbine load validation in wakes using field reconstruction techniques and nacelle lidar wind retrieval", by D. Conti, V. Pettas, N. Dimitrov, A. Penha.**

This manuscript proposes two methods for evaluating rotor loads under wake conditions. The work is primarily based on synthetic data generated through Mann's model and imposing modeled wake velocity deficits for different incoming wind speed and turbulence intensity.

I struggled to read through the entire manuscript and complete my review due to the cumbersome writing, lack of rigor of some statements and, sometimes, excessive technical details and jargons making more difficult the text comprehension. These are my main comments:
- In my opinion, this manuscript requires a major rewriting to sharpen its focus, remove jargons, and increase rigor in the description of the work.
- Many statements are not precise or incorrect, which makes the presentation of the work very cumbersome.
- This work uses a statistical approach to inject lidar data (here only simulated) in an existing velocity field through a technique proposed by the same authors in Dimitrov and Natarajan (2017). As shown in Figs. 4 and 12, this can produce reasonable characteristics of variance and spectra; however, it is far to be considered a data-assimilation technique (see more comments below). Maybe this method can be useful for wind energy applications, but it is highly below current standards for the turbulence/fluid mechanics community.

Please find below some comments, especially for the first three sections, which might help to rework the manuscript.

**Comments:**
1. The abstract should be sharpened to clarify the contribution of this manuscript and highlight the results achieved. There are too many details that result to be confusing without reading first the text in detail, see e.g., the "target observations", the baseline, etc.
2. L20, "*The wake-induced velocity deficit and its spatial displacement…*", just call it meandering.
3. L28, "*For the purpose of load validation, the IEC 61400-1 standard (IEC, 2019) recommends engineering wake models, which ensures low computational effort and an acceptable level of accuracy.*" This sentence can be rephrased. It sounds in contradiction with the previous paragraph. Maybe you can say that detailed predictions of wake-generated turbulence can be achieved with LES; however, the required computational cost makes engineering wake models a practical alternative.
4. L29, spell out DWM the first time in the text, even though you already mentioned it in the abstract.
5. L 54, "*wake deficit characteristics and their motions*": the motion of the wake deficit characteristics has no sense to me. Please clarify what you are trying to explain.
6. L56-59. Again, the description of the work is very confusing. If I am not mistaken, you compare the load predictions obtained with the two proposed models against those obtained by

injecting to the aeroelastic code more classical predictions obtained through the DWM model. Then, at L62 it is stated "*the load prediction obtained using lidar-reconstructed wake flow fields is as accurate or superior than that obtained with the DWM mode*". How can you get better accuracy of your benchmark dataset? At the very best, you can match those data with your new models.

7.      L60, "*two sets of independent turbulence seed realizations*", the meaning of this is not clear.

8.      L80-82. I disagree that you can quantify the statistical uncertainty of a turbulent process only by comparing two simulations. Furthermore, differences between the two simulations can be ascribed to both turbulence and wake meandering. How did you quantify the statistical distribution of your samples? How do you define the error between the two simulations? What statistical tests did you use to quantify the uncertainty?

9.      L83-84, "*we use a virtual lidar simulator that scans the target wake fields, and, through a field reconstruction technique, incorporates these samples in a random turbulence seed from set B*". This is quite an obscure description of your research! What field reconstruction technique? How do you incorporate samples from one simulation in the other one?

10.     L 103, The statistical uncertainty (i.e., standard deviation of the bias) ? I have never seen this definition of uncertainty. Provide references, if any.

11.     L 117, "wave vector with the wavenumbers in" at least remove wave.

12.     Sect 3.2 is a single paragraph with 20 lines, a great exercise for diving apnea training!

13.     L144-L148 and Fig. 1. You are presenting the results of simulations without providing any sort of basic description or references. For instance, how did you get the Ct of the turbine as a function of incoming wind speed, what incoming velocity did you use for the simulations with different turbulence intensity? What spatial resolution do you have in your data?

14.     Eq. 4, How did you select the standard deviation of the Gaussian weighting function? Why did you choose a Gaussian function to simulate the spatial averaging? Can you provide references? More realistic functions have been proposed in the past, see e.g., work by Mann.

15.     L156, this sentence "*The u-velocity is computed from the projection of VLOS,eq onto the longitudinal axis, i.e., the v- and w-velocity components are neglected in the field reconstruction*" is not correct unless you mention the constraints used in the angle difference between the velocity vector and the LOS vector. I guess we all agree that if the LOS vector is perpendicular to *u,* the LOS velocity is zero, but *u* is not.

16.     L 164, "*8192×32×32 (x,y,z)*" What is the corresponding spatial domain with respect to the used reference frame?

17.     L 166, "*These dimensions ensure an adequate turbulence field for a 10 min wind field simulation over a large rotor*" How did you assess this statement through the simulation data? Please add these details.

18.     L 171, maybe continuous wave (CW).

19.     L 170-188. This review of different lidars is not needed because this work is mainly numerical. Please remove this part and only describe the scanning strategy considered.

20.     L 196, what is a scan radius? Please define it.

21.     In Eq. 5 and 6, I guess you need to add time as an independent variable.

22.     L 219, "in Eq. (19) in Madsen et al. (2010)" I suggest to add this equation in the manuscript.

23.     L 223, The meaning of point 2 is unclear.

24.     L 224, What velocity fluctuations with reference to Eq. 5?

25.     L 226. Can you please define what are these turbulence seeds for set A and set B. To the best of my knowledge, turbulence seed is not mentioned in any turbulence book.

26.     L 245 "*that maintains the covariance and coherence properties of the unconstrained field ˜g(r)*" What about fulfilling the Navier-Stokes equations? Is this a real turbulent flow or only a collection of random numbers? Looking at Eqs. 7 and 8, I guess this is true for a random time-series. However, you cannot call these signals "turbulence". Other constraints and more sophisticated data-assimilation techniques should be considered to generate a turbulence field (see e.g., P. Bauweraerts, J. Meyers, J. Fluid Mech., Reconstruction of turbulent flow fields from lidar measurements using large-eddy simulation, 906, A17, 2020).

27.     There might be an inconsistency between Eq. 10 and Eq. 11., i.e., U_lidar=U_WDS? Furthermore, Eq. 11, states that Kdef,lidar is not only the imposed velocity deficit kdef, WDS with a random perturbation added $u'_B$. If that the case, then Eq. 11 is trivial and a simpler description can be provided.

28.     L 298, "explained variance"? This might be only acceptable as jargon among lab mates not for a scientific publication.

29.     L 373, what does is the list 8, 7, 7, 6,6,6 ,,,etc mean?

---

## Author Response (AR1)

**Author response to reviewer 1**

The authors response is shown in red

We thank the reviewer for the valuable comments and suggestions, which we consider very important and help us to sharpen and improve the manuscript. Here our response to each comment.

This paper presents methods for performing load validation by reconstructing the wind field upstream of a waked wind turbine using nacelle lidar measurements. The load validation methods are simulated using the dynamic wake meandering (DWM) model and aeroelastic simulations, and compared to the performance of the standard IEC recommended DWM method for load validation. The paper is a nice extension of previous work by the authors "Aeroelastic load validation in wake conditions using nacelle mounted lidar measurements," where the authors use wind parameters based on lidar measurements in wake conditions to evaluate the accuracy of load validation as part of a field experiment.

The paper is well written and clearly organized. Furthermore, the topic is relevant given the interest in using nacelle lidars for applications such as power performance and load validation in the wind industry. By investigating the proposed load validation methods in a controlled simulation environment, the authors are able to isolate the impact of the wind field reconstruction methodology, without worrying about aeroelastic model uncertainties. Although there are no major issues with the paper, there are several smaller comments that I believe should be addressed by the authors.

First, more motivation for the proposed lidar-based load validation methods should be presented. For example, if the goal is to achieve load prediction biases that are the same as the baseline method but with lower statistical uncertainty, how will this improve the wind turbine design process? And can you discuss current problems with the IEC-recommended approaches for load validation in wake conditions?

We added two paragraphs describing both the current limitations of the IEC-recommended approaches for load validation in wakes and how nacelle lidar-based procedures can tackle these issues. Further, we discuss the benefits of developing lidar-based power and load validation procedures in general.

Comments:
1. Title: Instead of "using field reconstruction techniques," which is somewhat vague, consider "using wind field reconstruction techniques"

This is now corrected.

2. Pg. 2, ln. 34: "the 10-min statistical properties (mean and variance) of the simulated ambient and operational conditions are set to match the measured ambient wind statistics": This doesn't quite make sense. How can the simulated "operational" conditions be set to match measured "ambient" conditions. Wouldn't you only need to match the ambient conditions?

This has been corrected, it is only the 'ambient' wind conditions that should be matched.

3. Pg. 2, ln 45: ": : :which increases the amount of validation data." This could use a little

more explanation (contrast this to a fixed met tower where only a small sector is valid).

We added more explanations to further clarify this.

4. Pg. 2, ln. 49: "The recent work of Conti et al. (2020) demonstrated that lidar-based load validation procedure in wakes should account for a model of the wake deficit and its dynamics." Since this paper builds on the work of Conti et al. 2020, please discuss this work in a little more detail, especially why it was concluded that lidar-based load validation in wakes should include a wake model.

We have added a paragraph discussing the main findings from the previous work.

5. Pg. 3, ln. 83: "through a field reconstruction technique": "wind field?" Or "wake field?"

'Wake field reconstruction technique' has been added.

6. Pg. 4, ln. 98: "large number of simulations" How many?

We added that we use 18 turbulence field realizations for each 10-min statistic of the inflow wind. Further, we specify that more details on the load validation analysis can be found in Sect. 4.2.

7. Pg. 4, ln. 101: "The mean bias of load predictions... is of the same order of that obtained with the baseline". Please be more specific about how close the lidar-based simulations should be to the baseline. "Of the same order" is a subjective criteria and makes it hard to tell if the new methods are successful.

We replaced "Of the same order" with "equal to".

8. Pg. 4, ln. 111: "the IEC recommends, i.e., the Mann uniform shear spectral tensor model..." This is one model that is recommended. There is also the Kaimal spectral model, etc.

The Kaimal spectral model has been added.

9. Eq. 1: the symbol "i" is used twice, for the spatial location as well as to indicate imaginary numbers. Can you choose unique symbols?

We now use $i$ for index and i for the imaginary numbers.

10. Eq. 2: Should the bold "k" argument on the left hand side be "$k_1$"?

This has been corrected.

11. Section 3.2: Can you explain more about the tools you are using to implement DWM? In other words, is DWM a software tool that you are using (if so, a reference would be appreciated)? Or is it a model described in the literature that you are implementing yourselves?

We have added more details and rephrased the whole subsection to better describe the assumptions of the DWM model. We also provide a reference of the numerical scheme of the DWM model

used to derive the quasi-steady velocity deficit in the paper.

12. Fig. 1: What wind speed is used for the middle plot?

We have removed this plot and replaced with a different figure, where we now define the inflow wind conditions.

13. Pg. 6, ln. 157: "The latter increases the uncertainty of the procedure." It is unclear what "the latter" refers to here.

We have now specified that in the text.

14. Pg. 6, ln. 166: "while the spatial resolution in the longitudinal axis depends on the simulated wind speed." Then what is the temporal resolution of the wind field?

The turbulence fields used in aeroelastic simulations (and in the DWM model) are basically a vector field, where each point in the field represents the local speed of the flow. In the generation of these fields we use the Taylor's assumption of frozen turbulence. Therefore, the large turbulence structures does not really change with time but are simply transported with the mean wind speed of the ambient wind field. As we run simulations with different ambient wind speeds, but the dimension of the turbulence box is fixed in the longitudinal axis to 8192 'points', the spatial resolution is function of $dx = (U_{amb}T_{sim})/8192$, where $T_{sim}$ is the simulation time in seconds (e.g., 600 s for a 10-min simulation).

15. Pg. 7, ln. 171: "continuous-wake" $->$ "continuous-wave"

This has been corrected.

16. Pg. 7, ln. 181: Can you provide a reference for the 4-beam Leosphere lidar?

This sentence has been removed together with the whole paragraph about the various type of nacelle lidars in the literature. For reference, the 4-beam Leosphere is advertised on the Leosphere's website.

17. Pg. 8, ln. 185: "7-beam lidar can potentially increase the accuracy of reconstructed wind fields." Increase the accuracy compared to what?

This sentence has been removed together with the whole paragraph about the various type of nacelle lidars in the literature.

18. Pg. 8, ln. 193-194: There is also a 4-beam Windar CW lidar, and the grid configuration pattern is based on the SWE pulsed lidar. Can you explain why you classified these scan patterns as pulsed and CW, respectively? Furthermore, since you are only modeling a single measurement range, it is unclear how you model CW and pulsed lidars any differently in you simulations. Can you explain this further? Lastly, you are giving up additional measurement points (and therefore potentially wind field reconstruction accuracy) by only using a single range for the pulsed lidars. Why didn't you use multiple range gates?

We have removed the paragraph describing the currently available nacelle lidars. The previous classification between CW and PL lidar was only made to reference the existing type of nacelle-lidars. Still, it did not influence the simulation results, as we mainly simulate the probe volume effects by a pre-defined weighting function. The reason for using a single range is conditional on the fact that we use DWM model-based fields as target fields. Indeed, the DWM model predicts quasi-steady wake deficits, which are computed according to a specified downstream distance. These deficits are meandered transversely, advected in stream-wise direction with the mean wind speed using Taylor's assumption, and superimposed on random turbulence field realizations (we have now described that in detail in Sect. 3.2). As the DWM model does not simulate turbulence evolution, we cannot simulate multiple range gates. This analysis would be suitable using an LES-based wake field. Another aspect to consider when using multiple ranges is that the wake recovers and expands with farther downstream distances; therefore, the wake field characteristics observed further up-stream of the rotor may be considerably different from those approaching the turbine rotor.

19. Pg. 8, ln. 199: "A preview distance of 0.7 D is assumed." In addition to the lidar mea-surement accuracy arguments, there seems to be an interesting dilemma when measuring the wake deficits upstream of a turbine. On one hand, I imagine you would want to measure close to the turbine to capture the true wake velocity deficit at the rotor plane. On the other hand, measuring too close will introduce induction zone effects. Can you discuss how you approached this issue?

We did not investigate this dilemma in detail in this work mainly because we use the DWM model-based fields as the target. As the DWM model does not include turbulence evolution and induction effects, we cannot investigate in detail what is an optimized preview distance for char-acterizing the inflow wind. In the work of [1], it is shown that an optimum preview distance for free-stream conditions varies between $0.4 - 1.3$ D according to specific lidar pattern and the specific wind field characteristics to be estimated. Here, we adopt a fixed preview distance of 0.7D, as we want to measure close to the rotor for the two reasons described in the text (i.e., reducing errors due to turbulence evolution and considering that lidar's probe volume typical increases for farther distances as for a continuous wave system). We have now added a few lines to discuss this.

20. Pg. 8, ln. 204: "A probe volume with an extension of 30 m in the LOS direction is assumed" Can you provide some references for how you chose 30 m for pulsed and CW lidars? Furthermore, how is the probe volume extension defined? For example, the std. dev. of Gaussian weighting function?

We have added that the probe volume length is here defined as the standard deviation of the Gaussian weighting function, and added references. The probe volume length of 30 m does not identify a specific lidar system, but it is an estimate that is comparable with the current CW lidar technology measuring at distances beyond 120 m [2]. Further, we conduct a sensitivity analysis by varying the probe volume lengths in Sect. 4.3.2, to analyze how these lengths influence the accuracy in power and load predictions.

21. Pg. 9, ln. 218: "obtained by simply scaling an isotropic turbulence field..." Can you clarify if the scaling depends on the radial location from the wake center, as shown in Fig. 1?

We have rephrased this sentence and provided a better description of the DWM model, includ-ing the wake-added turbulence formulation.

22. Pg. 9, ln. 221: How might the ambient wind conditions be measured in practice?

This is explained just a few lines below; see ln. 230. Ideally, from a met mast installed at the site or a nacelle lidar measuring the inflow wind.

23. Pg. 10, ln. 226: What do you mean by 'The u-velocity fluctuations are recovered from the target wake fields?'

We have rephrased to: 'Only the $u$-velocity fluctuations are reconstructed from the *target* wake fields.'

24. Pg. 11, ln. 256: "By denoting: : : as the constrained turbulence field that incorporates lidar measurements: : :" It seems that in Eq. 9, $u'_{CS,B,i}$ represents the turbulent fluctuations with the mean ambient wind profile removed. Do you first remove the mean ambient wind speeds from the lidar measurements before they are used to generate the constrained turbulence field?

That's correct. We have now described this step in the procedure.

25. Eq. 10: I'm confused about how $K_{def,lidar}$ is defined. From Fig. 1, $K_{def}$ is presented as a scaling factor applied to the ambient wind field ($= 1$, when wake losses are not present). But here, it appears to be defined as the normalized deficit ($= 0$, when wake losses are not present). Can you clarify this and make sure the definitions of $K_{def}$ are consistent?

That's correct, we now define $K_{def}$ as the normalized deficit ($= 0$, when wake losses are not present) and keep this definition consistently.

26. Eq. 10: Since the left hand side of this equation is being fit to the Gaussian function, they are not actually "equal." It would make more sense to present this equation as a minimization objective function (e.g., based on the difference between the measured deficit and the Gaussian model) Also, should $U_{amb(z)}$ have the mean operator applied to it, like in Eq. 9?

We have corrected the equation accordingly, and provided a minimization objective function instead.

27. Eqs. 10 and 11: The explanation of Eq. 11 is confusing. In your final method are you using the Gaussian fit from Eq. 10 as part of Eq. 11, or does Eq. 11 entirely replace Eq. 10? It would help to present both equations as minimization problems, so it's easy to see where the lidar measurements are being used and what exactly is being fit to the Gaussian profile.

We have corrected Equations 10 and 11 and provided a better description of the procedure.

28. Section 4.1: In addition to the analyses presented, a nice way to quantify the accuracy of the reconstructed wind fields could be to compare the RMSE of the rotor average wind speed $u_{eff}$ as well as the best-fit linear horizontal and vertical shear coefficient time series between the target and reconstructed wind fields. These variables should play a large role in determining the turbine loads.

We agree that it could also be an option. However, the analysis presented in Sect. 4.1 should provide sufficient information for evaluating how different lidar scanning configurations, complemented with the proposed wake field reconstruction techniques, perform. Further, we assess three wake field-related indicators ($U_{eff}$, $\rho_E^2$, $\sigma_u^2$) in the load validation analysis in Sect. 4.2, which should explain to a large extent, the observed deviations in power and load predictions.

29. Pg. 12, ln. 308: "run at the downstream distance of 5 D" Please be more specific. The turbine of interest is located 5 D downstream of the upstream turbine?

This has been corrected.

30. Fig. 7: On the left plot showing $U_{eff}/U_{amb}$, can you explain why the ratio converges to 0.93 at high wind speeds? As wind speed increases, the turbine thrust should keep decreasing causing wake losses to continue to decrease, so I would expect the ratio to approach 1.

It does not converge to 1 because although the trust coefficient decreases for higher wind speeds, the ambient turbulence is relatively low, and therefore the wake field does not fully recover at a distance of 5D, which is the one analyzed in this study. The ratio $U_{eff}/U_{amb}$ will converge to 1 for higher ambient turbulence or farther downstream distances due to the increased turbulence mixing. We have now described that in the paper.

31. Pg. 18, ln. 440: "In addition, improved estimates of both $\rho_E^2$ and $\sigma_u^2$ are seen in Fig. 10c,d." There seem to be improvements at low wind speeds, but slightly worse performance at high wind speeds. Can you comment on this in the paper?

We have now discussed this result in the paper.

32. Figs. 10 and 11: I would suggest full captions.

The full captions have been added.

33. Pg. 20, ln. 463: "focus the analysis on the SL, Grid, and Grid* configurations" These might be the most promising scan patterns, but also not the most likely, given currently available commercial lidar technology. It would be interesting to analyze the time series for one of the commercially-available lidar scenarios as well.

We opted to show only the most promising results, as the currently available commercial lidar technology (i.e., the 4P, 7P, and the Cone patterns) will introduce significant biases in the power and load predictions, as one can see from results in Figs. 8, 9, 10, and 11. This figure intends to show that provided a sufficient number of wind measurements taken upwind of the rotor, both the *CS*- and *WDS*-approach can reconstruct power and load time series that are highly correlated with the *target* observations. These results explain why we obtain lower statistical uncertainty $X_R$ values.

34. Pg. 21, ln. 469: "It should be noted that the structural resonance occurring at low wind speeds, which excites the tower can potentially affect the correlation results." Can you discuss why this resonance appears? Could it be removed by improving the controller tuning?

It appears because of the structural design of the DTU 10 MW, which is a reference (theoretical) turbine model. At low wind speeds (thus low RPM), the 3P rotational frequency (0.3–0.48 Hz) excites the eigenfrequency of the tower ($\approx 0.25$ Hz). Considering that the wake induces unbalanced

load distribution on the rotor, which in turn amplifies the rotor harmonics (1P, 2P, and 3P), this results in structural resonance. Besides that, we also observe that the bending moment of the tower bottom for large turbines is highly driven by the 3P frequency, as also shown in Fig. 13 (where the imprint of the turbulence wind is almost non-existence). Some internal work at DTU has been conducted to reduce the resonance, and the controller utilized in this work should be optimized to reduce resonance effects, which are still present and amplified under wake conditions. Future studies that evaluate these lidar-based reconstruction approaches can be conducted with different wind turbine designs that do not experience these resonances.

35. Pg. 22, ln. 481: Usually magnitude-squared coherence is written as $\gamma^2 = abs(S_x, y)^2/(S_x * S_y)$. Therefore, I would expect your definition to be $\gamma = abs(S_x, y)/sqrt(S_x * S_y)$. Is this correct?

This has been corrected as $\gamma^2 = |S(\tilde{y}, \hat{y})(f)|^2/(S(\tilde{y})(f)S(\hat{y})(f))$

36. Fig. 14: On the left plot, why is the baseline coherence so high at low frequencies (above the noise floor)?

This follows as the MxBR signal (blade root flapwise bending moment) is driven by both the wake meandering frequency and most importantly by the 1P rotational frequency (both are relatively low frequency signals as shown in Fig. 14). As the target and baseline simulations operates at similar 10-min average RPM (the ambient wind speed is the same), the 1P peak does not vary significantly between the Power Spectral Density of the target and the baseline MxBR loads. So we can see a non-zero coherence at low frequency, which is still lower than 0.3.

37. Pg. 25, ln. 538: As mentioned earlier, the "need for reducing the statistical load prediction uncertainty" in wake conditions could be motivated more clearly in the paper. More discussion or references talking about the need for improved methods would strengthen the message of the paper.

We have now addressed this point in the introduction, and delete this paragraph in the discussion section.

38. Pg. 26, ln. 562: You say that the lidar-based predicted load statistics are comparable to the results from the baseline DWM method ($\Delta_R$ between 0.97 - 1.01). However, from Figs. 9 and 11, it seems more accurate to say that $\Delta_R$ is between 0.92/0.94 and 1.01. Is 0.94 still an acceptable difference?

We have removed this paragraph from the discussions, as we discuss those results in the appropriate section (see Sect. 4.2). But, it is correct that $\Delta_R$ is between 0.92/0.94 and 1.01, depending on the load component, probe volume size and the adopted wake field reconstruction techniques.

39. Pg. 27, ln. 610: Similarly, the range of $\Delta_R$ with the lidar-based method is more like 0.94-1.01 instead of 0.97-1.01. When saying that this is comparable with the baseline method, please be more specific about what "comparable" means.

We have rephrased the conclusions accordingly.

40. Pg. 27, ln. 615: In addition to these lidar parameters, the load prediction accuracy is sensitive to the turbulence intensity as well.

This has been added.

41. Pg. 28, ln. 628: "largest energy content at higher frequencies ($> 3P$, 0.3 Hz)." From the plots, the largest energy content is at very low frequencies and right at 3P, but $> 3P$ does not contain as much energy content.

This has been corrected to "largest energy content at higher frequencies up to 3P (0.3 Hz)."

**References**

[1] Eric Simley, Holger Fürst, Florian Haizmann, and David Schlipf. Optimizing lidars for wind turbine control applications-results from the iea wind task 32 workshop. *Remote Sensing*, 10(6):863, 2018.

[2] Alfredo Peña, Charlotte Bay Hasager, Merete Badger, Rebecca Jane Barthelmie, Ferhat Bingöl, Jean-Pierre Cariou, Stefan Emeis, Sten Tronæs Frandsen, Michael Harris, Ioanna Karagali, Søren Ejling Larsen, Jakob Mann, Torben Mikkelsen, Mark Pitter, Sara Pryor, Ameya Sathe, David Schlipf, Chris Slinger, and Rozenn Wagner. Remote sensing for wind energy, 2015.

**Author response to reviewer 2**

The authors response is shown in red

We thank the reviewer for the valuable comments and suggestions, which we consider very important and help us to sharpen and improve the manuscript. Here our response to each comment.

This manuscript proposes two methods for evaluating rotor loads under wake conditions. The work is primarily based on synthetic data generated through Mann's model and imposing modeled wake velocity deficits for different incoming wind speed and turbulence intensity:

I struggled to read through the entire manuscript and complete my review due to the cumbersome writing, lack of rigor of some statements and, sometimes, excessive technical details and jargons making more difficult the text comprehension. These are my main comments:

- In my opinion, this manuscript requires a major rewriting to sharpen its focus, remove jargons, and increase rigor in the description of the work.

  A substantial update of the manuscript is carried out to clarify and sharpen the explanations. Further, an improved motivation of the study, a better description of the load validation procedure, and an improved description of the assumptions of the DWM model are provided to increase rigor in the description of the work.

- Many statements are not precise or incorrect, which makes the presentation of the work very cumbersome.

  The cumbersome statements highlighted by the reviewer have been improved or removed.

- This work uses a statistical approach to inject lidar data (here only simulated) in an existing velocity field through a technique proposed by the same authors in Dimitrov and Natarajan (2017). As shown in Figs. 4 and 12, this can produce reasonable characteristics of variance and spectra; however, it is far to be considered a data-assimilation technique (see more comments below). Maybe this method can be useful for wind energy applications, but it is highly below current standards for the turbulence/fluid mechanics community.

  The scope of the work (which has now been updated in the manuscript) is to verify that incorporating nacelle lidars measurements in the wake field reconstruction methods improve the accuracy of power and load predictions when compared to wake field reconstruction methods that are based on engineering wake models alone (e.g., the DWM model). The introduction section now motivates in detail the need for this study. From the improved manuscript, it should be now clear that the scope of this study is not to outperform data-assimilation techniques developed in the turbulence/fluid mechanics community, but to propose and demonstrate lidar-based techniques that are suitable and practical for engineering purposes such as power and load assessments under wake conditions at a given site, which require hundreds to thousands of aeroelastic simulations.

Comments:
1. The abstract should be sharpened to clarify the contribution of this manuscript and highlight the results achieved. There are too many details that result to be confusing without reading first the text in detail, see e.g., the "target observations", the baseline, etc.

   *The abstract has been sharpened by clarifying the main contributions and leaving details of the work outside.*

   2. L20, "The wake-induced velocity deficit and its spatial displacement...", just call it meandering.

   *This has been corrected.*

   3. L28, "For the purpose of load validation, the IEC 61400-1 standard (IEC, 2019) recommends engineering wake models, which ensures low computational effort and an acceptable level of accuracy." This sentence can be rephrased. It sounds in contradiction with the previous paragraph. Maybe you can say that detailed predictions of wake-generated turbulence can be achieved with LES; however, the required computational cost makes engineering wake models a practical alternative.

   *The sentence has been rephrased by emphasizing that as current state-of-the-art, LES can simulate wake flow fields accurately; however, they are still impractical in a design or site-specific power load assessment analysis.*

   4. L29, spell out DWM the first time in the text, even though you already mentioned it in the abstract.

   *This is now done.*

   5. L 54, "wake deficit characteristics and their motions": the motion of the wake deficit characteristics has no sense to me. Please clarify what you are trying to explain.

   *The sentence in L54 is unclear and has been corrected as: 'The second approach reconstructs wake deficit characteristics including wake meandering by fitting...'*

   6. L56-59. Again, the description of the work is very confusing. If I am not mistaken, you compare the load predictions obtained with the two proposed models against those obtained by injecting to the aeroelastic code more classical predictions obtained through the DWM model. Then, at L62 it is stated "the load prediction obtained using lidar-reconstructed wake flow fields is as accurate or superior than that obtained with the DWM model". How can you get better accuracy of your benchmark dataset? At the very best, you can match those data with your new models.

   *We have replaced that sentence with 'The main objective of this study is to verify that nacelle-mounted lidar measurements incorporated into wake field reconstruction methods improve the accuracy of power and load predictions when compared to wake field reconstruction using engineering wake models alone.'*
   *The sharpened introduction section clarifies better the limitations of the IEC-recommended engineering wake models for load calculations, and how lidar-based wake field reconstruction tech-*

niques can potentially tackle these limitations.

We have also improved Section 2 'Problem formulation', which provides a better description of the load validation procedure and how lidar-based wake field reconstruction methods can potentially outperform the DWM model by reducing the statistical uncertainty in power and load predictions. So, at very best the lidar-based wake field reconstruction approaches can fulfill the Criteria I and II described in Ln 101-104.

7. L60, "two sets of independent turbulence seed realizations", the meaning of this is not clear.

We have replaced the wording 'seed' with 'turbulence field'. A stochastic turbulence field generated with the Mann turbulence model or the Kaimal model (both are recommended in the IEC 61400-1 standard) is defined as a zero-mean homogeneous Gaussian turbulence field. Two random turbulence field realizations will produce two zero-mean homogeneous turbulence fields that are Gaussian, independent and uncorrelated (the realization of one turbulence field does not affect the probability distribution of the other).

8. L80-82. I disagree that you can quantify the statistical uncertainty of a turbulent process only by comparing two simulations. Furthermore, differences between the two simulations can be ascribed to both turbulence and wake meandering. How did you quantify the statistical distribution of your samples? How do you define the error between the two simulations? What statistical tests did you use to quantify the uncertainty?

The purpose of Section 2 'Problem formulation' is to formulate the load validation procedure and criteria used along the study. The exact details with regards to the questions: 'How did you quantify the statistical distribution of your samples? How do you define the error between the two simulations? What statistical tests did you use to quantify the uncertainty?' are defined and described in detail in 'Sect. 4.2 Load validation'. We provide a short description in here to answer the reviewer's comments:

We run a load validation analysis following the guidelines of the IEC 61400-13, which consists of applying a one-to-one comparison between predicted and measured (in our case *target*) power and load statistics (on a 10-min basis). This one-to-one load validation procedure is typically conducted in the design phase of a wind turbine to verify that the aeroelastic model predict loads accurately (see also IEC61400-13). Here, we extend this one-to-one load validation procedure under wake conditions in order to evaluate whether lidar-reconstructed wake fields, which are input to aeroelastic simulations, can predict power and loads accurately (e.g., with respect to the *target* results).

The IEC 61400-1 standard recommends using either the Mann model or the Kaimal model for generating random turbulence field realizations for aeroelastic simulations. Since these turbulence fields are stochastic, the resulting power and load predictions are affected by statistical uncertainty (e.g., load scatter). To overcome this issue, the IEC 61400-1 standard recommends performing aeroelastic simulations with at least 6 random turbulence field realizations for each 10-min realization of the inflow wind conditions, so to compute a more representative value of the loads. As described in Sect. 4.2, we use 18 turbulence field realizations (a factor of 3 higher than the recommendations of the IEC standard) for each 10-min realization of the inflow wind and quantify the statistical uncertainty in power and load predictions accordingly.

For example, given a wind speed of 6 m/s and $TI_{amb} = 6\%$, we generate 18 random turbulence field realizations that are input to the DWM model, run 18 aeroelastic simulations, and calculate the corresponding 18 values of the power and load statistics. We denote as bias the ratio between the simulated and *targeted* statistic for a single realization, and compute the uncertainty estimates

on the statistical distribution of this bias variable over multiple realizations. Therefore, we calculate the mean bias ($\Delta_R$ out of 18 simulations) and the standard deviation from all the 18 biases ($X_R$). In our study, the statistical uncertainty in power and load predictions presented in Figures 10 and 12 are computed out of 162 simulations and not only by comparing two simulations.

We agree that this procedure does not describe all the statistical uncertainty of a turbulent process; however, this is not our goal. The goal is to describe the statistical uncertainty due to the differences between the simulated and targeted power and load predictions inherent to traditional load validation procedures (i.e., the realization-to-realization uncertainty).

'Differences between the two simulations can be ascribed to both turbulence and wake meandering'. This is correct, and we have discussed this point in the introduction (Ln 34–40). Indeed, this study's primary purpose is to verify that incorporating nacelle lidar measurements in the wake field reconstruction methods improves the accuracy and decrease the uncertainty in wake field representations (and consequently that of power and load fluctuations). How? As one can reduce the statistical uncertainty occurring due to the stochastic nature of the turbulence fields and the wake meandering time series (among others) that are inherent to conventional engineering wake models (such as the DWM model).

'How did you quantify the statistical distribution of your samples?' See Ln 372. According to the IEC recommendations at least 6 random turbulence field realizations should be used to account for statistical uncertainty in power and load predictions; here we use 18 turbulence field realizations to ensure we can accurately estimate the statistical uncertainty.

'How do you define the error between the two simulations?' See Ln. 351-356 and the whole Sect. 4.2, i.e., using $\Delta_R$ and $X_R$ indicators.

'What statistical tests did you use to quantify the uncertainty?' See answers to comment nr. 10 below.

9. L83-84, "we use a virtual lidar simulator that scans the target wake fields, and, through a field reconstruction technique, incorporates these samples in a random turbulence seed from set B". This is quite an obscure description of your research! What field reconstruction technique? How do you incorporate samples from one simulation in the other one?

This sentence belongs to a section (Sect. 2 'Problem formulation') that explains what field reconstruction techniques are used in the study. We have corrected the sentence with 'and through our proposed wake field reconstruction techniques,...'. The proposed approaches are explained few lines below (Ln. 90-95) as well as in the abstract (Ln. 1-6) and introduction (Ln. 51-56). Further, Sects. 3.4.1 and 3.4.2. describe how we incorporate lidar samples into the wake field reconstruction methods.

10. L 103, The statistical uncertainty (i.e., standard deviation of the bias) ? I have never seen this definition of uncertainty. Provide references, if any.

We have rephrased it with 'The statistical uncertainty (here defined as the standard deviation computed from all biases (Dimitrov et al. (2017) and Conti et al. (2020)) ...'. We show an illustrative example in Fig. 1 and then provide a detailed description.

Our study defines two uncertainty indicators to assess the power and load predictions accuracy: $\Delta_R$ and $X_R$ (see also Fig. 1-right), which are defined mathematically in Sect. '4.2 Load validation' Ln. 350-352 together with a detailed description of the performed aeroelastic simulations. Indeed, for each 10-min realization of the inflow wind conditions (e.g., given a wind speed of 6 m/s

[Figure]

[Figure]

Figure 1: Left: Probability distribution function (PDF) of the mean power productions ($Power_{mean}$) obtained from 100 aeroelastic simulations (thus 100 random turbulence field realizations) with an inflow wind speed of 6 m/s and $TI_{amb} = 6\%$. Right: The $Power_{mean}$ results are normalized with respect to the *target* results. The mean and standard deviation values (i.e., $\mu$ and $\sigma$ that corresponds to $\Delta_R$ and $X_R$ when normalized with the *target* results) are reported in the figure.

and $TI_{amb}=6\%$), we run aeroelastic simulations with 18 random turbulence field realizations, and quantify the mean bias between predicted and *target* load statistics ($\Delta_R$), and a measure of the standard deviation of these biases (out of 18 values) that is $X_R$ (i.e., the load's scatter dispersion).

Since the standard deviation is typically used as a measure of uncertainty in model predictions, here we use $X_R$ that is mathematically defined as the standard deviation of the bias. (Figure 1 should clarify this, and we show results from 100 simulations for illustrative purpose only).

We use $\Delta_R$ and $X_R$ as we aim to verify the load validation criteria described in 'Sect. 2 Problem formulation': (I) evaluating that lidar-reconstructed wake field provides unbiased power and load predictions. (II) verifying that the statistical uncertainty (which is here quantified using $X_R$ that is the standard deviation of the biases computed out of all the simulations) is lower when using lidar-based wake fields than conventional DWM model-based fields.

11. L 117, "wave vector with the wavenumbers in" at least remove wave.

This has been removed.

12. Sect 3.2 is a single paragraph with 20 lines, a great exercise for diving apnea training!

This section has been divided into smaller paragraphs. Further, we have rephrased the text to clarify the underlying flow modeling assumptions of the DWM model.

13. L144-L148 and Fig. 1. You are presenting the results of simulations without providing any sort of basic description or references. For instance, how did you get the Ct of the turbine as a function of incoming wind speed, what incoming velocity did you use for the simulations with different turbulence intensity? What spatial resolution do you have in your data?

We agree that this figure lacks essential information. This figure's purpose was to provide a qualitative illustration of how the wake deficit recovers for increasing ambient turbulence and wind speeds. As we improved the description of the DWM model in Sect. 3.2, we have also replaced this figure.

14. Eq. 4, How did you select the standard deviation of the Gaussian weighting function? Why did you choose a Gaussian function to simulate the spatial averaging? Can you provide references? More realistic functions have been proposed in the past, see e.g., work by Mann.

The weighting function of a continuous-wake lidar is often approximated by a Lorentzian form [1]. However, the Gaussian weighting approximation may also be used [2, 3]. Dimitrov et al. (2019) [3] quantified a difference in the $u$-velocity variance of less than 3% when using a Gaussian weighting function compared to the Lorentzian form.

For the data sets used in the present study, the difference between using a Gaussian and a Lorentzian weighting function was negligible. Figure 2 shows a comparison between the Gaussian- and Lorentzian-like weighting functions. As shown, using a Gaussian function has negligible effects when reconstructing the $U$-velocity component (longitudinal velocity component that is the primary driver to power and load predictions). The procedure to derive the $U$-velocity component is provided below.

[Figure]

Figure 2: Lidar's weighting functions (i.e., Lorentzian and Gaussian shape). $D$ denotes the rotor diameter of the DTU 10 MW wind turbine, $Z_r$ is the probe volume length defined as the standard deviation of the Gaussian function or as the half-width-half-maximum for the Lorentzian function. $U$ defines the reconstructed velocity component accounting for the weighting function as described in the text below. An estimate of the resulting $U$ velocity is provided in the plot accounting for both the Lorentzian and Gaussian weighting functions.

The procedure to derive the $U$-velocity component: as the lidar simulator scans numerical wind fields, the sampled data points are discrete, and therefore the velocity estimates represent the

weighted sum of a distribution of velocity measurements along the line-of-sight $(v_{los})$ as:

$$\tilde{v}_{los} = \frac{\sum_{i=1}^{n_p} \varphi(s_i) v_{los}(s_i)}{\sum_{i=1}^{n_p} \varphi(s_i)}, \tag{1}$$

where $n_p$ indicates the number of discrete points along the measurement volume. The weighting function $(\varphi(s_i))$ is approximated by a Gaussian function. The line-of-sight velocity is then expressed as function of the wind speed components and the geometrical angle $(\phi, \theta)$, where $\phi$ is the elevation and $\theta$ the azimuth angle:

$$\tilde{v}_{los}(\phi, \theta) = u \cos \phi \cos \theta + v \cos \phi \sin \theta + w \sin \phi. \tag{2}$$

Yet the wind field at a given location cannot be fully characterized using a single lidar, instead a retrieval assumption is required to characterize the longitudinal velocity component, which is the major driver to power and load calculations. Considering that $u \gg v, w$, the measured radial wind speed is typically presumed to be due to the $u$ component alone with $v=w=0$. Thus Eq. (2) becomes:

$$\tilde{v}_{los}(\phi, \theta) = U \cos(\phi) \cos(\theta). \tag{3}$$

Equation (3) allows the virtual lidar simulator to reconstruct the horizontal wind velocity at each individual scanned point within the scanning configuration.

15. L156, this sentence "The u-velocity is computed from the projection of VLOS,eq onto the longitudinal axis, i.e., the v- and w-velocity components are neglected in the field reconstruction" is not correct unless you mention the constraints used in the angle difference between the velocity vector and the LOS vector. I guess we all agree that if the LOS vector is perpendicular to u, the LOS velocity is zero, but u is not.

This is correct; however, the maximum opening angles relative to the scanning configurations analyzed in this work reach a maximum of 35°. Further, for large opening angles (e.g., larger than $\approx 25°$), the lidar is actually measuring an area that is outside of the rotor area. Thus, the uncertainty introduced by the flow assumptions ($v$ and $w=0$) is marginal, and it is anyway discussed as one of the sources of uncertainty that affect the accuracy in power and load predictions.

16. L 164, "8192×32×32 (x,y,z)" What is the corresponding spatial domain with respect to the used reference frame?

As described in Ln. 165: 'A spatial resolution of 6.5 m is used for the grid in the rotor plane, which leads to a turbulence box with dimension 208 m × 208 m in both lateral and vertical directions $(y, z)$.'.
Note that the turbulence fields used in aeroelastic simulations (and in the DWM model) are vector fields, where each grid point represents the local speed of the flow. In the generation of these fields, we use Taylor's assumption of frozen turbulence. Therefore, the large turbulence structures do not really change with time but are simply transported with the mean wind speed of the ambient wind field. As we run simulations with different ambient wind speeds, but the dimension of the turbulence box is fixed in the longitudinal axis to 8192 'points', the spatial resolution is function of $dx = (U_{amb} T_{sim})/8192$, where $T_{sim}$ is the simulation time in seconds (e.g., 600 s for a 10-min simulation). We have added this to the paper.

17. L 166, "These dimensions ensure an adequate turbulence field for a 10 min wind field simulation over a large rotor" How did you assess this statement through the simulation data? Please add these details.

A turbulence field with 32x32 points is considered sufficient because the field is internally downsampled to approximately 15 points per blade when running the Blade Element Momentum (BEM) code in the HAWC2 software [4]. Larger turbulence boxes can be used, but they will not affect the result's accuracy but only increase the storage required to generate larger turbulence boxes. As we generate over 1000 simulations, we opted to keep the computational and storage requirements low without compromising the results' accuracy.
We also added a reference to Dimitrov and Natarajan (2017), who used the DTU 10 MW for load validation analysis and found these dimensions to be suitable for load calculations.

18. L 171, maybe continuous wave (CW).

This has been corrected.

19. L 170-188. This review of different lidars is not needed because this work is mainly numerical. Please remove this part and only describe the scanning strategy considered.

This part has been removed.

20. L 196, what is a scan radius? Please define it.

We added a definition as: 'we use scan radii (defined as the radius between hub height and the location of the scanned points)...'

21. In Eq. 5 and 6, I guess you need to add time as an independent variable.

The DWM model assumes Taylor's frozen turbulence hypothesis; therefore, the wind field is described by the spatial vector solely. We have added a line in the text to describe this. Further, 'Sect. 3.2 Dynamic Wake Meandering model' has been improved to describe better the DWM model's assumptions.

22. L 219, "in Eq. (19) in Madsen et al. (2010)" I suggest to add this equation in the manuscript.

This has been added

23. L 223, The meaning of point 2 is unclear.

We have slightly rephrased point 2 as: 'The lidar-based wake fields are reconstructed by incorporating lidar observations (e.g., in the form of constraints or lidar-fitted velocity deficits) into a zero-mean, homogeneous, and random Gaussian turbulence field generated by the Mann spectral tensor model.'
We state this assumption as the lidar-based wake field reconstruction methods are similar to the wake field reconstruction methods inherent in engineering wake models. Indeed, the wake features are either pre-computed using a physical-based model (e.g., the DWM model) or fitted through lidar data (e.g., the CS and WDS algorithms of the present work). Successively, these lidar-based

or physical-based wake features are superposed on stochastic homogeneous turbulence fields generated by the Mann model. By doing so, we keep the computational time as that for engineering wake models; thus, the lidar-based techniques are practical for power and load validation analyses, which require many aeroelastic simulations.

As described in a previous comment (see 13.), we have now rephrased 'Sect. 3.2 Dynamic Wake Meandering model' to emphasize the underlying assumptions of the DWM model, so Point 2 becomes more evident to the reader.

24. L 224, What velocity fluctuations with reference to Eq. 5?

If this comment refers to what velocity fluctuations are reconstructed by the lidar-based wind field reconstruction procedures, then Sect. 3.4.1 and 3.4.2 should clarify this. We have also added an equation that relates the LOS velocity to the u-velocity component.

25. L 226. Can you please define what are these turbulence seeds for set A and set B. To the best of my knowledge, turbulence seed is not mentioned in any turbulence book.

We agree that 'seed' is not appropriate here. We have replaced the term 'seed' with 'turbulence field realization' throughout the whole paper. The seed method is used to initialize the random number generator to create a random turbulence field realization from the Mann turbulence model.

26. L 245 "that maintains the covariance and coherence properties of the unconstrained field $\tilde{g}(r)$ What about fulfilling the Navier-Stokes equations? Is this a real turbulent flow or only a collection of random numbers? Looking at Eqs. 7 and 8, I guess this is true for a random timeseries. However, you cannot call these signals "turbulence". Other constraints and more sophisticated data-assimilation techniques should be considered to generate a turbulence field (see e.g., P. Bauweraerts, J. Meyers, J. Fluid Mech., Reconstruction of turbulent flow fields from lidar measurements using large-eddy simulation, 906, A17, 2020).

The Mann model is used in this work because it describes the atmospheric-turbulence velocity spectra for different surface, wind, and atmospheric-stability conditions (see i.a., [5, 6]). Further, the Mann model is recommended in the IEC 61400-1 and -3 standards for modeling three-dimensional turbulence fields required as input to aeroelastic simulations, and is widely used in load validation analysis (see i.a., [3, 7])

We agree that more sophisticated data-assimilation techniques exist (e.g., the work of P. Bauweraerts, J. Meyers, J. Fluid Mech., Reconstruction of turbulent flow fields from lidar measurements using large-eddy simulation, 906, A17, 2020). We have cited and discussed it in this paper's discussion section (Ln. 545-551). However, these high-fidelity techniques are yet not practical for power and load assessments that require many aeroelastic simulations (i.e., hundreds to thousands).

We have also sharpened the work scope to clarify that we are not aiming at outperforming LES-based data-assimilation techniques but providing a practical alternative to engineering wake model-based power and load assessment procedures commonly used in the wind energy industry today.

27. There might be an inconsistency between Eq. 10 and Eq. 11., i.e., $U_{lidar} = U_{WDS}$? Furthermore, Eq. 11, states that Kdef,lidar is not only the imposed velocity deficit kdef, WDS with a random perturbation added u'B. If that the case, then Eq. 11 is trivial and a simpler description can be provided.

This has been corrected and a simpler description is now provided.

28. L 298, "explained variance"? This might be only acceptable as jargon among lab mates not for a scientific publication.

The explained variance is actually used as a statistical term, e.g. [Achen, C. (1982) Interpreting and Using Regression, Sage Publications] [8]. In its classical use it is defined as the proportion of the variance in the dependent variables which can be accounted for by a mathematical model. For a regression model this is equivalent to the coefficient of determination (the square of the Pearson's correlation coefficient). Since in this work we use the definition in the broader sense, we prefer to retain the term explained variance. We have rephrased this in the text, so to clarify that we did not self-defined it.

29. L 373, what does is the list 8, 7, 7, 6,6,6 ,,,etc mean?

The list indicates the corresponding turbulence intensity values for each analyzed wind speed ranging from 6 to 22 m/s with a 2 m/s step. We have rephrased the text so it is clearer.

[revised manuscript text omitted]
=$ 10~~MW wind turbine, as function of ambient wind speed and turbulence.As illustrated, the depth of the wake deficit decreases for increasing inflow wind speed and turbulence , due to the reduced rotor thrust coefficient and the enhanced turbulence mixing. The wake-added turbulence is proportional to the depth and gradient of the velocity deficit profile; thus a faster recovery of the deficit induces lower added turbulence . Steady wake characteristics predicted by the DWM model at the downstream distance of 5 D, D being the rotor diameter, resolved in the radial direction and normalized on the rotor radius (R = 89.5 m). Left and middle: wake deficit profiles as function of ambient wind speed and ambient turbulence. Right: wake-added turbulence profiles as function of the ambient turbulence.~~ % 
[revised manuscript text omitted]

355 ~~calculated with the DWM model, and $u'_i$ is a random turbulence realization from the Mann model with spectral properties as for the ambient wind field. The spectral properties of the turbulence are defined by $\alpha_k\epsilon^{2/3}, L, \Gamma$, which can be fitted based on freestream observations. The wind field formulation of Eqs. and is consistent with the domain of wind fields typically input to aeroelastic simulations. Finally, $u'_{j,K_{turb}}$ is obtained by simply scaling an isotropic turbulence field with low turbulence length scales, by the semi-empirical formulation in Eq. (19) in Madsen et al. (2010).*target*~~ 
[revised manuscript text omitted]

785 ~~reconstruct to a large extent the wake dynamics that have the strongest impact on the power and load predictions. These include the spatial distribution of the velocity deficit and the added turbulence resulting from its motions in time. The lidar-based predicted load statistics are comparable to the results obtained with the IEC-recommended DWM model ($\Delta_R \sim 0.97 - 1.01$). Furthermore, the statistical uncertainty of the lidar-based load predictions is considerably reduced by a factor between 1.2–5 compared to DWM's results (the $baseline$). By combining lidar measurements with the $CS$- and $WDS$-reconstruction~~

790

 Characterizing the small-scale wake-added turbulence poses a challenge given the  limitations of lidar's sampling frequency and probe volume size (Peña et al., 2017). The  small-scale wake-added turbulence enhances the energy spectral content in the high-frequency range,  0.4–20 Hz (Madsen et al., 2010;

795 Chamorro et al., 2012; Singh et al., 2014), and its contribution on the fatigue damage varies according to the load component and turbine operational strategy (Tibaldi et al., 2015).

Bergami and Gaunaa (2014) demonstrated that the most serious fatigue damage on the blades occurs at frequencies around 1P (0.1–0.16 Hz for the DTU 10 MW), whereas structures as tower top (nacelle) and tower bottom are mainly affected by the tower eigenfrequency ($\approx$ 0.25 Hz) and the 3P frequency (0.3–0.48 Hz). As the PSD of tower loads exhibits large energy

800 spectral content at high frequencies (see Fig. A1), the accuracy of tower load predictions decreases compared to that achieved by blade loads, as  found in Sects. 4.2.4 and 4.2.5.

We demonstrate that a high  number of lidar-scanned positions of the inflow is required to ensure an acceptable level of accuracy  in the reconstructed wake fields. The results reveal that the current commercially available nacelle-mounted lidars  (e.g., the 4P, 7P, and Cone patterns) will not provide

805 sufficient information to  reconstruct the wake fields  accurately for the load assessments. In contrast, the scanning requirements are  fulfilled by the SpinnerLidar and any arbitrary lidar that can potentially scan a greater region of the rotor,  e.g., a Grid-like configuration. Although we do not optimize the scanning strategies, it is inferred that the required  number of positions scanned by the lidar depends on the size of the  turbulence structures in the  wake field.

810  Incorporating a sufficient number of lidar measurements directly in the turbulence  fields leads to more accurate load predictions ~~, compared to assuming a generic functional shape of the wake deficit. This is shown by the time-series analysis in Sect. 4.2.4 and the spectral coherence analysis of Sect. 4.2.5, which illustrate that high frequency load fluctuations can be reconstructed more accurately by the $CS$- than the $WDS$-approach. Furthermore, the constrained field technique based on the Mann turbulence model can be extended to incorporate~~ 
[revised manuscript text omitted]

Bos, R., Giyanani, A., and Bierbooms, W.: Assessing the severity of wind gusts with lidar, Remote Sensing, 8, 758, https://doi.org/10.3390/rs8090758, 2016.

Bossanyi, E.: Un-freezing the turbulence: application to LiDAR-assisted wind turbine control, Iet Renewable Power Generation, 7, 321–329, https://doi.org/10.1049/iet-rpg.2012.0260, 2013.

Bossanyi, E. A., Kumar, A., and Hugues-Salas, O.: Wind turbine control applications of turbine-mounted LIDAR, Journal of Physics: Conference Series, 555, 012011, https://doi.org/10.1088/1742-6596/555/1/012011, 2014.

Chamorro, L. P., Guala, M., Arndt, R. E., and Sotiropoulos, F.: On the evolution of turbulent scales in the wake of a wind turbine model, Journal of Turbulence, 13, 1–13, https://doi.org/10.1080/14685248.2012.697169, 2012.

Churchfield, M. J., Moriarty, P. J., Hao, Y., Lackner, M. A., Barthelmie, R., Lundquist, J. K., and Oxley, G. S.: A comparison of the dynamic wake meandering model, large-eddy simulation, and field data at the egmond aan Zee offshore wind plant, 33rd Wind Energy Symposium, pp. 20 pp., 20 pp., 2015.

Conti, D., Dimitrov, N., and Peña, A.: Aero-elastic load validation in wake conditions using nacelle-mounted lidar measurements, Wind Energy Science Discussions, 2020, 1–31, https://doi.org/10.5194/wes-2020-8, https://www.wind-energ-sci-discuss.net/wes-2020-8/, 2020a.

Conti, D., Dimitrov, N. K., and Peña, A.: Aeroelastic load validation in wake conditions using nacelle-mounted lidar measurements, Wind Energy Science, 5, 1129–1154, https://doi.org/10.5194/wes-5-1129-2020, 2020b.

de Mare, M. T. and Mann, J.: On the Space-Time Structure of Sheared Turbulence, Boundary-layer Meteorology, 160, 453–474, https://doi.org/10.1007/s10546-016-0143-z, 2016.

Dimitrov, N., Borraccino, A., Peña, A., Natarajan, A., and Mann, J.: Wind turbine load validation using lidar-based wind retrievals, Wind Energy, 22, 1512–1533, https://doi.org/10.1002/we.2385, 2019.

Dimitrov, N. K. and Natarajan, A.: Application of simulated lidar scanning patterns to constrained Gaussian turbulence fields for load validation, Wind Energy, 20, 79–95, https://doi.org/10.1002/we.1992, 2017.

Dimitrov, N. K., Natarajan, A., and Mann, J.: Effects of normal and extreme turbulence spectral parameters on wind turbine loads, Renewable Energy, 101, 1180–1193, https://doi.org/10.1016/j.renene.2016.10.001, 2017.

Dimitrov, N. K., Kelly, M. C., Vignaroli, A., and Berg, J.: From wind to loads: wind turbine site-specific load estimation with surrogate models trained on high-fidelity load databases, Wind Energy Science, 3, 767–790, https://doi.org/10.5194/wes-3-767-2018, 2018.

Doubrawa, P., Barthelmie, R. J., Wang, H., Pryor, S. C., and Churchfield, M. J.: Wind turbinewake characterization from temporally disjunct 3-D measurements, Remote Sensing, 8, 939, https://doi.org/10.3390/rs8110939, 2016.

[revised manuscript text omitted]

---

## Author Response (AR2)

**Author response to reviewer 1**

The authors response is shown in red

We thank the reviewer for the valuable comments and suggestions, which we consider very important and help us to sharpen and improve the manuscript. Here our response to each comment.

The paper has been improved greatly. It now contains stronger motivation for the work, clearer explanations of the methods, and an improved summary of the results. Given that this is a long paper and there were many changes made to it, I do have many mostly minor comments that I believe should be addressed for the final version.

Comments on Author Responses:

18. Pg. 8, ln. 193-194: There is also a 4-beam Windar CW lidar, and the grid configuration pattern is based on the SWE pulsed lidar. Can you explain why you classified these scan patterns as pulsed and CW, respectively? Furthermore, since you are only modeling a single measurement range, it is unclear how you model CW and pulsed lidars any differently in you simulations. Can you explain this further? Lastly, you are giving up additional measurement points (and therefore potentially wind field reconstruction accuracy) by only using a single range for the pulsed lidars. Why didn't you use multiple range gates?

AR: We have removed the paragraph describing the currently available nacelle lidars. The previous classification between CW and PL lidar was only made to reference the existing type of nacelle lidars. Still, it did not influence the simulation results, as we mainly simulate the probe volume effects by a pre-defined weighting function. The reason for using a single range is conditional on the fact that we use DWM model-based fields as target fields. Indeed, the DWM model predicts quasi-steady wake deficits, which are computed according to a specified downstream distance. These deficits are meandered transversely, advected in stream-wise direction with the mean wind speed using Taylor's assumption, and superimposed on random turbulence field realizations (we have now described that in detail in Sect. 3.2). As the DWM model does not simulate turbulence evolution, we cannot simulate multiple range gates. This analysis would be suitable using an LES-based wake field. Another aspect to consider when using multiple ranges is that the wake recovers and expands with farther downstream distances; therefore, the wake field characteristics observed further upstream of the rotor may be considerably different from those approaching the turbine rotor.

"As the DWM model does not simulate turbulence evolution, we cannot simulate multiple range gates": The simplification to one range is fine for the paper. But it would still be possible to simulate multiple ranges without wind evolution. This is done frequently when assessing lidar-assisted control. It just adds an additional assumption about the wind field.

As the reviewer said, it is correct that multiple ranges could be simulated with the DWM model while neglecting wind evolution. However, we did not analyze multiple ranges as part of this work.

Also, since many readers will be familiar with pulsed and CW lidars, please discuss the assumptions used in the paper (e.g., that you are modeling a pulsed lidar at one range, a CW lidar, or that your model is not specific to one type of lidar).

Since we removed the whole paragraph, there is no specification on the type of lidars. In order

to clarify that our model is not specific to one type of lidar, we now added two lines in section 3.3.1. Ln. 261, as: "The lidar simulator is assumed to scan the selected patterns at the same single range upwind of the rotor. Pulsed and continuous-wave (CW) lidar technologies apply different approaches at scanning multiple ranges. Pulsed lidars can scan multiple ranges along the LOS simultaneously within a single sample, while CW lidars typically sample much faster at a given range but need to refocus in order to change the sampling range. In the present paper, we only consider a single focusing range that is achievable with both lidar technologies. Further, a time lag between each sampling beam is simulated to mimic lidars' sampling frequency. "

20. Pg. 8, ln. 204: "A probe volume with an extension of 30 m in the LOS direction is assumed" Can you provide some references for how you chose 30 m for pulsed and CW lidars? Furthermore, how is the probe volume extension defined? For example, the std. dev. of Gaussian weighting function?

AR: We have added that the probe volume length is here defined as the standard deviation of the Gaussian weighting function, and added references. The probe volume length of 30 m does not identify a specific lidar system, but it is an estimate that is comparable with the current CW lidar technology measuring at distances beyond 120 m [2]. Further, we conduct a sensitivity analysis by varying the probe volume lengths in Sect. 4.3.2, to analyze how these lengths influence the accuracy in power and load predictions.

"but it is an estimate that is comparable with the current CW lidar technology measuring at distances beyond 120 m [2]": This is a reasonable simplification, but I would provide an explanation like this in the paper. Further, a 30 m probe length is commonly used to model pulsed lidars, so that might be a better justification to use. For example, 30 m is used as the full-width-at-half-maximum probe volume in:

Schlipf, D. Lidar-Assisted Control Concepts for Wind Turbines. Ph.D. Thesis, University of Stuttgart, Stuttgart, Germany, 2016.

We have rephrased it as: "A probe volume with an extension of 30 m in the LOS direction is assumed for all the analyzed patterns, which is comparable with the current continuous-wave lidar technology measuring at distances beyond 120 m (Peña et al., 2015). Further, a 30 m probe length is commonly used to model pulsed lidars (Schlipf 2016) [1].".

23. Pg. 10, ln. 226: What do you mean by 'The u-velocity fluctuations are recovered from the target wake fields?'

AR: We have rephrased to: 'Only the u-velocity fluctuations are reconstructed from the target wake fields.'

Consider "...are reconstructed from the lidar measurements of the target wake fields."

This has been corrected.

25. Eq. 10: I'm confused about how $K_{def,lidar}$ is defined. From Fig. 1, $K_{def}$ is presented as a scaling factor applied to the ambient wind field ($= 1$, when wake losses are not present). But here, it appears to be defined as the normalized deficit ($= 0$, when wake losses are not present). Can you clarify this and make sure the definitions of $K_{def}$ are consistent?

AR: That's correct, we now define $K_{def}$ as the normalized deficit ($= 0$, when wake losses are not present) and keep this definition consistently.

This is clear now in Sect. 3.4.2. However, the definitions of $U_{def}$ and $K_{def}$ in Eqs. 13+14 do not appear to be consistent with the definitions in Fig. 2 and Eq. 6. I.e., in Eqs. 13+14, $U_{def}$ and

$K_{def}$ are defined as 0 outside of the wake deficit region. But in Fig. 2 and Eq. 6 it appears they both equal 1 outside of the wake region.

That's correct. For consistency, we have corrected Eq. 6 to: $u'_{i,K_{def}}(x,y,z) = \bar{U}_{amb}(z)(1 - K_{def}(x,y,z)) + u'_i(x,y,z) - \bar{U}_{amb}(z)$. Further, we re-plotted Fig. 2, so $U_{def}= 0$, when wake losses are not present.

30. Fig. 7: On the left plot showing $U_{eff}/U_{amb}$, can you explain why the ratio converges to 0.93 at high wind speeds? As wind speed increases, the turbine thrust should keep decreasing causing wake losses to continue to decrease, so I would expect the ratio to approach 1.

AR: It does not converge to 1 because although the trust coefficient decreases for higher wind speeds, the ambient turbulence is relatively low, and therefore the wake field does not fully recover at a distance of 5D, which is the one analyzed in this study. The ratio $U_{eff}/U_{amb}$ will converge to 1 for higher ambient turbulence or farther downstream distances due to the increased turbulence mixing. We have now described that in the paper.

The lower turbulence at higher wind speeds does explain part of why the wake would not recover as much as expected by 22 m/s. However, since the ratio plateaus at 0.93 for several wind speed bins, it seems like something else is happening. Is $U_{amb}$ treated as the mean freestream wind speed at hub height? If that is the case, then maybe even in freestream conditions, $U_{eff}$ will be $0.93 * U_{amb}$ because of wind shear.

We checked the simulations' results, and the contribution of the wind shear to the ratio $U_{eff}/U_{amb}$ should account for up to 3% in free-stream conditions ($U_{eff}/U_{amb} \approx 0.97$). Another aspect influencing the $U_{eff}/U_{amb}$'s trend at high wind speeds (18–22 m/s) is that the thrust coefficient does not vary with the same rate compared to below rated wind speeds but it is nearly constant.

We have rephrased the text as: "However, the wake deficit does not fully recover at high wind speeds ($U_{eff}/U_{amb} \approx 0.93$), as we simulate relatively low ambient turbulence levels, the spacing between the turbines is short (i.e., 5 D), and the thrust coefficient of the turbine is nearly constant at high wind speeds. Further, the contribution of the wind shear to the ratio $U_{eff}/U_{amb}$ accounts for up to 3% in free-stream conditions, i.e., $U_{eff}/U_{amb} \approx 0.97$."

34. Pg. 21, ln. 469: "It should be noted that the structural resonance occurring at low wind speeds, which excites the tower can potentially affect the correlation results." Can you discuss why this resonance appears? Could it be removed by improving the controller tuning?

AR: It appears because of the structural design of the DTU 10 MW, which is a reference (theoretical) turbine model. At low wind speeds (thus low RPM), the 3P rotational frequency (0.30.48 Hz) excites the eigenfrequency of the tower ( 0.25 Hz). Considering that the wake induces unbalanced load distribution on the rotor, which in turn amplifies the rotor harmonics (1P, 2P, and 3P), this results in structural resonance. Besides that, we also observe that the bending moment of the tower bottom for large turbines is highly driven by the 3P frequency, as also shown in Fig. 13 (where the imprint of the turbulence wind is almost non-existence). Some internal work at DTU has been conducted to reduce the resonance, and the controller utilized in this work should be optimized to reduce resonance effects, which are still present and amplified under wake conditions. Future studies that evaluate these lidar-based reconstruction approaches can be conducted with different wind turbine designs that do not experience these resonances.

A sentence about the cause of the controller resonance would be insightful in the paper.

We have rephrased the sentence in page 21 as: "The $X_R$ values of MyTB and MzSh are significantly higher than other load sensors. The cause of the former is structural resonance occurring at low wind speeds for which the 3P frequency ($\approx 0.3$ Hz) excites the tower's natural frequency ($\approx 0.25$ Hz) (Back 2013). This effect originates from a design aspect of the DTU 10 MW turbine, and is amplified under wake conditions due to the induced unbalanced aerodynamic load distribution at the rotor. Nevertheless, structural resonance is independent of the wake-field reconstructing approach."

Additional Comments:

1. Ln. 55: "Further, to accurately reconstruct wake meandering time series, it is essential to ensure accurate power and load predictions in a load validation analysis"? This seems to make more sense the other way around: "to accurately predict power and loads in a load validation analysis, it is essential to accurately reconstruct wake meandering time series." Is this correct?

This is correct. We replaced the sentence.

2. Lns. 74-77: Would "monitoring wind turbine performance" make more sense as "condition monitoring of wind turbines"? Additionally, brief examples of how lidar-based power and load validation under wakes would improve the listed application areas would be appreciated.

We have rephrased the paragraph in Lns. 74-77 as: "Overall, developing lidar-based wake wind field reconstruction techniques that reduce the modeling and statistical uncertainties in the inflow inherent of low-order engineering wake models can improve loads and lifetime estimations accuracy (Rommel et al.,2020), enhance power curve testing in wind farms (Lydia et al., 2014; Wagner et al., 2015), and promote lidar-assisted wind turbine and wind farm control strategies (Bossanyi et al., 2014; Raach et al., 2017; Simley et al., 2018; Schlipf et al., 2020).". We added two references regarding lidar-based control strategies that use wake-tracking (Raach et al., 2017 [2]) and turbulence estimation (Schlipf et al., 2020 [3]) as input to the controller.

3. Fig. 2: In the middle plot, $k_{mt}$ appears to be 1 outside of the wake deficit region. But if this represents wake-added turbulence, should $k_{mt}$ be zero outside of the wake region?

This is correct. We have replaced the figure accordingly.

4. Ln. 197: "... scales the residual field of a Mann-generated turbulence field". What is the TI or std. dev. of the turbulence field. i.e., if $k_{mt}$ is used to scale the turbulence, then what is the baseline turbulence level that it is scaling?

This information was missing. We added that the std. of the turbulence field is 1 m/s, as described in the IEC standard [4].

5. Lns. 221-222: "from fitting the free-stream observed turbulence velocity spectra with the Mann model with the use of pre-computed look-up-tables". It isn't clear how look-up-tables would be used for this.

The look-up-tables (LUT) are used to efficiently compute the Mann model spectra ($F_{uu}, F_{vv},$

$F_{ww}, F_{uw}$) given the Mann parameters $(L, \Gamma, \alpha\epsilon^{2/3})$. Indeed, a bivariate spline approximation is carried out to determine the spectra from the LUT instead of analytically computing the spectra. The LUT approach is useful when extracting Mann parameters through an optimization procedure.

We added an explanation in the manuscript, which specifies that LUT is used to speed-up the fitting procedure.

6. Lns. 226-228: Similarly, what is the std. dev. of the $u'_j$ time series?

We added that $\sigma_u = 1$ m/s [4] .

7. Eq. 9: How are the elevation and azimuth angle defined? If azimuth is defined as the azimuth angle in the rotor plane (similar to azimuth angle of a blade), it is hard to see why the cos(theta) appears in the estimate of $u_{lidar}$.

We have rephrased the sentence as: "where $\phi$ is the elevation and $\theta$ the azimuth angle of the scanning pattern, which refer to the rotations about the $y$ and $z$ axes, respectively." The $y$ and $z$ axis are defined in Sect. 3.1 and also shown in Fig. 2.

8. Lns. 275-279: Since Gaussian weighting functions are typically used to model pulsed lidars and Lorentzian functions are used to model CW lidars, I would mention this point in the paper.

We have now mentioned it in the paragraph.

9. Section 3.4: Is it correct that the high-frequency wake added turbulence is not explicitly included in the 2 wake field reconstruction methods? I wasn't sure while reading the section, so it might be good to highlight this point.

Theoretically, the *CS* algorithm can reconstruct the high-frequency wake added turbulence. However, this would require sampling the wind field at a high temporal frequency (e.g., $f_{sampling} >$ 6 Hz as seen in Fig. 7) without probe volume effects. In contrast, the *WDS*-method cannot explicitly reconstruct the high-frequency wake-added turbulence. We added two sentences in the discussion of the manuscript to highlight this aspect, as in Sect. 3.4, we haven't yet introduced the methods.

10. Ln. 290: "set A and set B": It might be good to remind the reader that set A is used for the target fields.

We now added that in parenthesis.

11. Ln. 309: The constraint set is hard to understand. For example, what is the dimension of $H$? Should "$r$" be "$r_i$" in the definition of $H$, if the constraint is for a specific location? Finally, if each constraint is a measured time series, then should $c_i$ be written as $c_i(t)$? And is $M$ the number of points in the scan pattern?

We corrected the notation to $\boldsymbol{H} = \{h_i(\boldsymbol{r})|_{r_i} = c_i, i, ..., M\}$, following the notation of Dimitrov et al. 2017 [5]. Each constraint is a measured value of the wind speed for a particular spatial location $\boldsymbol{r}$, but not at time-series; we corrected the text accordingly. $M$ is the number of scanned points within a 10-min period, and we now specify that in the text.

12. Ln. 377: "The normalized RMSE indicates if the lidar-reconstructed fields are unbiased compared..." How would RMSE indicate the bias? The mean error would indicate bias, whereas RMSE could be caused by variability in the error.

That's correct. We have rephrased it as: "The normalized RMSE provides a measure of the quality of the lidar-reconstructed fields with respect to the *target* fields; values closer to zero indicate a high precision and accuracy (see Fig. 5-top row)."

13. Ln. 415: "These effects are not fully recovered in the reconstructed fields, mainly due to the lidar probe volume..." Also because the method fits the lidar measurements to a standard Mann turbulence field, without the small-scale wake-added turbulence being explicitly included.

This has been added.

14. Ln. 540: "underpredicted by $\Delta_R$ 2-3%". Based on Fig. 12 the bias can be up to 6%.

This has been corrected.

15. Ln. 554: Is it accurate to call the power time series the "$Power_mean$" time-series? Mean would suggest the mean over the 10-minute period, but you are looking at the full time series, correct?

This is correct, we have replaced "Power-mean" with "Power".

16. Lns. 606-608: "This indicates that when L is low,..." In addition, the turbulence structure sizes become small relative to the lidar probe volume, causing the lidar measurements to average out more of the turbulence.

This is correct and it has been added.

Minor Comments:

1. Ln. 73: "power and load" $->$ "power and loads"

This has been corrected.

2. Ln. 451: "2 m/s" $->$ "2 m/s bin width"? Also, consider adding "respectively" at the end of the sentence.

This has been corrected, and 'respectively' is added at the end of the sentence.

3. Ln. 491: "40-60% estimates" $->$ "40-60% accuracy"?

This has been corrected.

4. Ln. 537: "fictitious biases" $->$ "fictitious lack of biases"?

This has been corrected.

**References**

[1] David Schlipf. *Lidar-assisted control concepts for wind turbines*. PhD thesis, 2016.

[2] Steffen Raach, David Schlipf, and Po Wen Cheng. Lidar-based wake tracking for closed-loop wind farm control. *Wind Energy Science*, 2(1):257–267, 2017.

[3] David Schlipf, Feng Guo, and Steffen Raach. Lidar-based estimation of turbulence intensity for controller scheduling. *Journal of Physics: Conference Series*, 1618(3):032053, 2020.

[4] International Standard IEC61400-1: wind turbines—part 1: design guidelines, Fourth; 2019. Standard, IEC, 2019.

[5] Nikolay Krasimirov Dimitrov and Anand Natarajan. Application of simulated lidar scanning patterns to constrained gaussian turbulence fields for load validation. *Wind Energy*, 20(1):79–95, 2017.

**Author response to reviewer 2**

The authors response is shown in red

We thank the reviewer for the valuable comments and suggestions, which we consider very important and help us to sharpen and improve the manuscript. Here our response to each comment.

The manuscript seems improved from the first version. Here two additional comments:

• Fig. 2 does not add any crucial information for the manuscript and could be removed, together with the respective text (L220 on the marked-up version).

We agree that the figure does not add any crucial information for the manuscript; however, we decided to keep it as it facilitates the understanding of the DWM model for those readers who are not familiar with the model.

• Cross-check the second term in Eq. 4

This has been corrected.

**References**